# Linear Convergence of Gradient Methods for Estimating Structured Transition Matrices in High-dimensional Vector Autoregressive Models

**Xiao Lv, Wei Cui**
School of Information and Electronics
Beijing Institute of Technology
{xiaolv, cuiwei}@bit.edu.cn

**Yulong Liu**\*
School of Physics
Beijing Institute of Technology
yulongliu@bit.edu.cn

## Abstract

In this paper, we present non-asymptotic optimization guarantees of gradient descent methods for estimating structured transition matrices in high-dimensional vector autoregressive (VAR) models. We adopt the projected gradient descent (PGD) for single-structured transition matrices and the alternating projected gradient descent (AltPGD) for superposition-structured ones. Our analysis demonstrates that both gradient algorithms converge linearly to the statistical error even though the strong convexity of the objective function is absent under the high-dimensional settings. Moreover our result is sharp (up to a constant factor) in the sense of matching the phase transition theory of the corresponding model with independent samples. To the best of our knowledge, this analysis constitutes first non-asymptotic optimization guarantees of the linear rate for regularized estimation in high-dimensional VAR models. Numerical results are provided to support our theoretical analysis.

## 1 Introduction

Learning the network structure through high-dimensional time series data has been an important focus of research for the past decades. There are lots of application examples ranging from macroeconomic analysis [1–3] to connectivity measuring among financial firms [4], gene regularity network inference [5] and radar signals processing [6, 7]. To perform these tasks, vector autoregressive (VAR) models play a critical role in both theory and application. For example, VAR models are widely adopted to characterize the spatially and temporally colored disturbance for multichannel adaptive signal detection in [7–11].

Under the low-dimensional settings where the dimension of the transition matrix and the number of time series are relatively small, the theory of VAR models is well established, see e.g., [12]. However, lots of meaningful applications are under the high-dimensional settings where the problem dimension far exceeds the number of time series and additional structure information of parameters is required to guarantee successful recovery. For the cases where the samples are independent, both theoretical properties and practical algorithms of high-dimensional statistical problems have been studied by considerable literature during the past few years, including but not limited to [13–17]. For the correlated time series cases, the corresponding results are still developing. By assuming that the spectral norm of the transition matrix is less than 1, Loh and Wainwright consider VAR models regularized by the $l_1$-norm in [18]. Under the double asymptotic framework, Han and Liu also consider VAR models under the similar assumption in [19]. In [20], Basu and Michailidis analyze sparse transition matrices estimation of VAR models with a milder stability assumption by

---

\*Corresponding author.

35th Conference on Neural Information Processing Systems (NeurIPS 2021).

introducing the spectral density. Recently, Melnyk and Banerjee extend the analysis to structured VAR models regularized by any suitable norm in [21].

Compared with the progress on the statistical analysis of VAR models, much less is known about their computational issues. For instance, Basu et al. [22] make exploration in this direction by proposing the fast network structure learning algorithm (FNSL) for the penalized recovery procedure. Their analysis does not take the structure information of the parameter into account and it only establishes a sub-linear convergence rate for the FNSL.

In this paper, we first provide the non-asymptotic optimization guarantee of PGD for VAR models with single-structured transition matrices. Our analysis illustrates that the distance between iteration points of PGD and the real transition matrix would converge linearly to the statistical error despite the objective function is not strongly convex under the high-dimensional settings. Our result is sharp in the sense that the minimal requirement of samples to guarantee the linear rate matches the phase transition of the model with independent samples up to a constant factor.

On the other hand, considering the parameter to be estimated only has one type of low-dimensional structure or is single-structured might be too oversimplified for messy real applications. So superposition-structured models have received more attention of researchers in last decade. Typical examples include robust PCA [23, 24], multi-task learning [25] and robust matrix sensing [26]. For these scenarios, we employ AltPGD to solve related optimization problems and establish the corresponding non-asymptotic optimization guarantee. Our results show that AltPGD also enjoys a linear convergence rate, which is much more efficient than the sub-linear rate in [22]. At the same time, our analysis avoids a drawback in [22] that the estimation error does not converge to zero when the number of measurements approaches infinity.

Last but not the least, we also illustrate that AltPGD is a practical algorithm to solve the general superposition-structured statistical model in [27]. Apart from the time series case, our analysis also adapts to multi-task learning and robust PCA.

## 2   Problem formulation

In this paper, we consider a $d$-dimensional vector-valued stationary time series $\{\boldsymbol{x}_0, \cdots, \boldsymbol{x}_n\}$ generated by a VAR model of lag 1 with serially uncorrelated Gaussian errors. The VAR(1) model is defined as

$$\boldsymbol{x}_{t+1} = \boldsymbol{\Gamma}_\star^T \boldsymbol{x}_t + \boldsymbol{e}_{t+1}, \qquad t = 0, \cdots, n-1, \tag{1}$$

where $\boldsymbol{\Gamma}_\star \in \mathbb{R}^{d \times d}$ is the transition matrix and $\boldsymbol{e}_t \overset{iid}{\sim} \mathcal{N}(\boldsymbol{0}, \boldsymbol{\Sigma}_e)$. This model can be reformulated in the matrix form

$$\boldsymbol{Y} = \boldsymbol{X}\boldsymbol{\Gamma}_\star + \boldsymbol{E}, \tag{2}$$

where $\boldsymbol{Y} = [\boldsymbol{x}_1, \cdots, \boldsymbol{x}_n]^T \in \mathbb{R}^{n \times d}$, $\boldsymbol{X} = [\boldsymbol{x}_0, \cdots, \boldsymbol{x}_{n-1}]^T \in \mathbb{R}^{n \times d}$, and $\boldsymbol{E} = [\boldsymbol{e}_1, \cdots, \boldsymbol{e}_n]^T \in \mathbb{R}^{n \times d}$.

The goal of the VAR(1) model is to recover the transition matrix $\boldsymbol{\Gamma}_\star$ from the observation matrix $\boldsymbol{Y}$ and the data matrix $\boldsymbol{X}$. In the high-dimensional settings with $n \ll d^2$, tractable recovery is possible when the transition matrix $\boldsymbol{\Gamma}_\star$ is well structured. Thus we introduce a convex regularizer $\mathcal{R}(\cdot)$ to promote the structure of $\boldsymbol{\Gamma}_\star$. Then a popular way to estimate $\boldsymbol{\Gamma}_\star$ is to solve the following constrained least square problem

$$\begin{aligned} \min_{\boldsymbol{\Gamma}} \quad & \frac{1}{2n}\|\boldsymbol{Y} - \boldsymbol{X}\boldsymbol{\Gamma}\|_{\mathrm{F}}^2, \\ \text{s.t.} \quad & \mathcal{R}(\boldsymbol{\Gamma}) \leq \mathcal{R}(\boldsymbol{\Gamma}_\star), \end{aligned} \tag{3}$$

where $\|\cdot\|_{\mathrm{F}}$ represents the Frobenius norm of a matrix.

When the transition matrix $\boldsymbol{\Gamma}_\star$ is superposition-structured in which $\boldsymbol{\Gamma}_\star$ is the sum of two single-structured components, i.e., $\boldsymbol{\Gamma}_\star = \boldsymbol{S}_\star + \boldsymbol{L}_\star$, we adopt two convex functions $\mathcal{R}_S(\cdot)$ and $\mathcal{R}_L(\cdot)$ to characterize the structures of its two components and solve the following constrained problem to

estimate $\boldsymbol{S}_\star$ and $\boldsymbol{L}_\star$

$$
\begin{aligned}
\min_{\boldsymbol{S},\boldsymbol{L}} \quad & \frac{1}{2n}\|\boldsymbol{Y} - \boldsymbol{X}(\boldsymbol{S} + \boldsymbol{L})\|_{\mathrm{F}}^2, \\
\text{s.t.} \quad & \mathcal{R}_S(\boldsymbol{S}) \leq \mathcal{R}_S(\boldsymbol{S}_\star), \\
& \mathcal{R}_L(\boldsymbol{L}) \leq \mathcal{R}_L(\boldsymbol{L}_\star).
\end{aligned}
\tag{4}
$$

## 3 Single-structured transition matrices estimation via PGD

In this part, we consider the case where the transition matrix to be estimated in (1) only has one type of low-dimensional structure which is characterized by a convex function $\mathcal{R}(\cdot)$. To estimate the transition matrix $\boldsymbol{\Gamma}_\star$, we solve the problem (3) via the PGD update (summarized in Algorithm 1).

By setting $f_n(\boldsymbol{\Gamma}) = \|\boldsymbol{Y} - \boldsymbol{X}\boldsymbol{\Gamma}\|_{\mathrm{F}}^2/(2n)$, we could write the iteration of PGD as

$$
\boldsymbol{\Gamma}_{k+1} = \mathcal{P}_{\mathcal{K}}(\boldsymbol{\Gamma}_k - \mu\nabla f_n(\boldsymbol{\Gamma}_k)),
\tag{5}
$$

where $\mu$ is the step size, $\mathcal{K} = \{\boldsymbol{\Gamma} \mid \mathcal{R}(\boldsymbol{\Gamma}) \leq \mathcal{R}(\boldsymbol{\Gamma}_\star)\}$ is the descent set and $\mathcal{P}_{\mathcal{K}}$ represents the orthogonal projection onto the set $\mathcal{K}$. We also introduce the concept of the descent cone $\mathcal{C} = \mathrm{cone}(\mathcal{K} - \boldsymbol{\Gamma}_\star)$, where $\mathrm{cone}(\cdot)$ is the conic hull of a set.

---

**Algorithm 1** PGD for single-structured transition matrices estimation

**Input:** Initial point $\boldsymbol{\Gamma}_0$, step size $\mu$, iteration number $K$.
**for** $k = 0$ **to** $K - 1$ **do**
$\quad \boldsymbol{\Gamma}_{k+1} = \mathcal{P}_{\mathcal{K}}(\boldsymbol{\Gamma}_k - \mu\nabla f_n(\boldsymbol{\Gamma}_k))$
**end for**
**Output:** $\boldsymbol{\Gamma}_K$

---

To guarantee consistent estimation, we propose the following stability assumption for the VAR model (1), which is also imposed in [20, 22].

**Assumption 1** (Stability). *The characteristic polynomial of the VAR model* (1) *satisfies* $\det(\mathcal{A}(z)) \neq 0$ *on the unit circle of the complex plane* $\{z \in \mathbb{C}\colon |z| = 1\}$, *where* $\mathcal{A}(z) = \boldsymbol{I}_{d\times d} - \boldsymbol{\Gamma}_\star^T z$.

In the non-asymptotic analysis of the VAR model (1), we use the following two quantities

$$
\mathcal{M}(f_x) = \operatorname*{ess\,sup}_{\theta\in[-\pi,\pi]} \lambda_{\max}(f_x(\theta)),
\tag{6}
$$

$$
m(f_x) = \operatorname*{ess\,inf}_{\theta\in[-\pi,\pi]} \lambda_{\min}(f_x(\theta)),
\tag{7}
$$

where $f_x(\theta)$ is the spectral density function defined as

$$
f_x(\theta) \coloneqq \frac{1}{2\pi}\sum_{l=-\infty}^{\infty} \boldsymbol{\Sigma}_x(l)e^{-il\theta}, \qquad \theta \in [-\pi, \pi].
\tag{8}
$$

Here we use $\boldsymbol{\Sigma}_x(l)$ to represent

$$
\boldsymbol{\Sigma}_x(l) = \mathbb{E}[\boldsymbol{x}_t\boldsymbol{x}_{t+l}^T], \qquad t, l \in \mathbb{Z}.
\tag{9}
$$

Specially, we write $\boldsymbol{\Sigma}_x(0) = \boldsymbol{\Sigma}_x$ for simplicity.

Compared with the model with independent samples, there is dependency among the rows of data matrix $\boldsymbol{X}$ in the VAR model (2), which is the main challenge when deriving the deviation bounds required by the estimation problem. For Gaussian processes, this dependency could be characterized by the covariance matrix $\boldsymbol{\Upsilon}_x = \mathbb{E}[\mathrm{vec}(\boldsymbol{X}^T)\mathrm{vec}(\boldsymbol{X}^T)^T]$, where $\mathrm{vec}(\cdot)$ represents the column-wise vectorization of a matrix. The following lemma indicates that the concept of the spectral density function is a convenient tool to bound the extreme eigenvalues of $\boldsymbol{\Upsilon}_x$.

**Lemma 1** (Proposition 2.3 in [20]). *Donate* $\boldsymbol{\Upsilon}_x = \mathbb{E}[vec(\boldsymbol{X}^T)vec(\boldsymbol{X}^T)^T]$, *where* $\boldsymbol{X}$ *is the data matrix in the VAR model* (2). *We could bound the extreme eigenvalues of* $\boldsymbol{\Upsilon}_x$ *as*

$$
2\pi m(f_x) \leq \lambda_{\min}(\boldsymbol{\Upsilon}_x) \leq \lambda_{\max}(\boldsymbol{\Upsilon}_x) \leq 2\pi\mathcal{M}(f_x).
\tag{10}
$$

*In particular, we also have*

$$
2\pi m(f_x) \leq \lambda_{\min}(\boldsymbol{\Sigma}_x) \leq \lambda_{\max}(\boldsymbol{\Sigma}_x) \leq 2\pi\mathcal{M}(f_x).
\tag{11}
$$

For stable and invertible ARMA processes which include the model (1), the spectral density (8) has a closed form expression based on the matrix valued polynomials [20, Equation (2.4)]. Furthermore, the concrete calculation of the upper bound of $\mathcal{M}(f_x)$ and the lower bound of $m(f_x)$ for the model (1) is provided in [20, Proposition 2.2] which indicates $m(f_x)$ and $\mathcal{M}(f_x)$ could be bounded away from zero and infinity. In this way, we could introduce the quantities $\kappa_{\min}$ and $\kappa_{\max}$ to simplify the expression of our analysis.

**Assumption 2** (Boundness). *Suppose there are positive constants $\kappa_{\min}$ and $\kappa_{\max}$ satisfying*

$$0 < \frac{\kappa_{\min}}{2\pi} \leq m(f_x) \leq \mathcal{M}(f_x) \leq \frac{\kappa_{\max}}{2\pi}. \tag{12}$$

With the above assumption, we could represent the extreme eigenvalues of $\boldsymbol{\Upsilon}_x$ and $\boldsymbol{\Sigma}_x$ in a concise way

$$\kappa_{\min} \leq \lambda_{\min}(\boldsymbol{\Upsilon}_x) \leq \lambda_{\max}(\boldsymbol{\Upsilon}_x) \leq \kappa_{\max}, \tag{13}$$
$$\kappa_{\min} \leq \lambda_{\min}(\boldsymbol{\Sigma}_x) \leq \lambda_{\max}(\boldsymbol{\Sigma}_x) \leq \kappa_{\max}. \tag{14}$$

In our analysis, we use the Gaussian width to quantify the size of a set $\mathcal{T}$

$$\omega(\mathcal{T}) := \mathbb{E}\sup_{\boldsymbol{x} \in \mathcal{T}} \langle \boldsymbol{g}, \boldsymbol{x} \rangle, \qquad \text{where } \boldsymbol{g} \sim \mathcal{N}(\boldsymbol{0}, \boldsymbol{I}).$$

We are now ready to present the non-asymptotic optimization guarantee of PGD for the problem (3).

**Theorem 1.** *Consider the VAR model (1) satisfying Assumptions 1 and 2. Suppose $\boldsymbol{\Gamma}_\star$ is single-structured and $\mathcal{R}(\cdot)$ is a convex function. Starting from a point $\boldsymbol{\Gamma}_0$ satisfying $\mathcal{R}(\boldsymbol{\Gamma}_0) \leq \mathcal{R}(\boldsymbol{\Gamma}_\star)$, we solve the optimization problem (3) via PGD with the step size $\mu = 1/\kappa_{\max}$. If the number of measurements satisfies*

$$\sqrt{n} > 2C\frac{\kappa_{\max}}{\kappa_{\min}}(\omega(\mathcal{C} \cap \mathbb{S}_F) + u), \tag{15}$$

*then the PGD update (5) would obey*

$$\|\boldsymbol{\Gamma}_{k+1} - \boldsymbol{\Gamma}_\star\|_{\mathrm{F}} \leq \rho^{k+1}\|\boldsymbol{\Gamma}_0 - \boldsymbol{\Gamma}_\star\|_{\mathrm{F}} + \frac{\xi}{1-\rho} \tag{16}$$

*with probability at least $1 - c\exp(-u^2)$. Here*

$$\rho = 1 - \frac{\kappa_{\min}}{\kappa_{\max}} + C\frac{\omega(\mathcal{C} \cap \mathbb{S}_F) + u}{\sqrt{n}} < 1 - \frac{\kappa_{\min}}{2\kappa_{\max}}, \tag{17}$$

$$\xi = C'\frac{1}{\sqrt{\kappa_{\max}}}\|\boldsymbol{\Sigma}_e\|^{\frac{1}{2}}\frac{\omega(\mathcal{C} \cap \mathbb{S}_F) + u}{\sqrt{n}}, \tag{18}$$

$$\frac{\xi}{1-\rho} = \frac{1}{1-\rho} \cdot C'\frac{1}{\sqrt{\kappa_{\max}}}\|\boldsymbol{\Sigma}_e\|^{\frac{1}{2}}\frac{\omega(\mathcal{C} \cap \mathbb{S}_F) + u}{\sqrt{n}} < 2C'\frac{\sqrt{\kappa_{\max}}}{\kappa_{\min}}\|\boldsymbol{\Sigma}_e\|^{\frac{1}{2}}\frac{\omega(\mathcal{C} \cap \mathbb{S}_F) + u}{\sqrt{n}}, \tag{19}$$

*$\|\cdot\|$ represents the spectral norm of a matrix, $\mathbb{S}_F$ represents the sphere with unit Frobenius norm and $c, C, C'$ are absolute constants.*

**Remark 1** (Sharpness). *Our result demonstrates that PGD can converge linearly to the statistical error despite the objective function $f_n(\boldsymbol{\Gamma})$ is not strongly convex under the high-dimensional settings. And the linear convergence is achieved when the number of measurements is of order $\omega(\mathcal{C} \cap \mathbb{S}_F)^2$, which matches the phase transition of the model with independent samples [15, 16].*

**Remark 2** (Impact of correlated samples). *Our result also provide some insights for the impact of correlated samples. It is not hard to find that the temporal dependency is characterized by $\kappa_{\max}$ and $\kappa_{\min}$ which appear in the convergence rate, the estimation error, and the required number of samples. Let $\kappa = \kappa_{\max}/\kappa_{\min}$. Clearly, a smaller $\kappa$ will lead to a faster convergence rate with smaller required samples and estimation error.*

**Remark 3** (Comparison with related works). *The existing works on structured VAR models such as [19–21] are mainly concerned with establishing the statistical error bounds for different recovery procedures. Our work focuses on algorithmic analysis and reveals how many samples would ensure that the algorithm achieves a fast convergence rate. On the other hand, our work could be regarded*

*as a generalization of the result in [17] from independent samples to time series. This generalization is nontrivial, since new mathematical tools (e.g., two deviation inequalities: Lemmas 4 and 5) are required to address time-series settings. To the best of our knowledge, Lemma 5 seems to be the first non-asymptotic result for the VAR models which yields the unified estimation error bounds for both independent and correlated samples. Furthermore, our analysis demonstrates PGD enjoys a linear convergence for structured VAR models, which is much more efficient than the sub-linear convergence rate of FNSL in [22]. Additionally, the linear convergence rate of PGD is also illustrated in [18] for the VAR(1) model with a sparse transition matrix, when the spectral norm of the transition matrix is strictly less than 1. Compared with the result in [18], our analysis adapts to general structured signals with the milder Assumption 1.*

**Remark 4** (Extension). *In Theorem 1, we set the step size $\mu = 1/\kappa_{\max}$ for a concise expression of the result. In fact, any step size satisfying $\mu \leq 1/\kappa_{\max}$ could achieve a linear convergence rate by providing the corresponding number of measurements. In [20, 21], VAR(d) models are reformulated as VAR(1) models. With the same reformulation, our analysis also adapts to VAR(d) models.*

Our analysis for the VAR model (1) also adapts to the multi-task learning problem with independent samples.[1] Different from the time series setting, the measurements $\boldsymbol{Y} = \boldsymbol{X}\boldsymbol{\Gamma}_\star + \boldsymbol{E}$ in this case are generated from $\boldsymbol{x}_t \overset{iid}{\sim} \mathcal{N}(\boldsymbol{0}, \boldsymbol{\Sigma}_x)$ and $\boldsymbol{e}_t \overset{iid}{\sim} \mathcal{N}(\boldsymbol{0}, \boldsymbol{\Sigma}_e)$, for $t = 1, \cdots, n$, where $\boldsymbol{X} = [\boldsymbol{x}_1, \cdots, \boldsymbol{x}_n]^T \in \mathbb{R}^{n \times d}$ and $\boldsymbol{E} = [\boldsymbol{e}_1, \cdots, \boldsymbol{e}_n]^T \in \mathbb{R}^{n \times d}$ are independent.

**Corollary 1.** *Consider the multi-task learning problem with the above conditions. Under Assumption 2, we solve the optimization problem (3) via PGD with the step size $\mu = 1/\kappa_{\max}$ and a starting point $\boldsymbol{\Gamma}_0$ satisfying $\mathcal{R}(\boldsymbol{\Gamma}_0) \leq \mathcal{R}(\boldsymbol{\Gamma}_\star)$. If the number of measurements satisfies*

$$\sqrt{n} > 2C \frac{\kappa_{\max}}{\kappa_{\min}} (\omega(\mathcal{C} \cap \mathbb{S}_F) + u), \tag{20}$$

*then the update (5) would obey*

$$\|\boldsymbol{\Gamma}_{k+1} - \boldsymbol{\Gamma}_\star\|_F < (1 - \frac{\kappa_{\min}}{2\kappa_{\max}})^{k+1} \|\boldsymbol{\Gamma}_0 - \boldsymbol{\Gamma}_\star\|_F + 2C' \frac{\sqrt{\kappa_{\max}}}{\kappa_{\min}} \|\boldsymbol{\Sigma}_e\|^{\frac{1}{2}} \frac{\omega(\mathcal{C} \cap \mathbb{S}_F) + u}{\sqrt{n}} \tag{21}$$

*with probability at least $1 - c\exp(-u^2)$, where $c$, $C$, $C'$ are absolute constants.*

## 4 Superposition-structured transition matrices estimation with AltPGD

In this part, we consider the case where the transition matrix $\boldsymbol{\Gamma}_\star$ to be estimated in (1) is superposition-structured, that is, $\boldsymbol{\Gamma}_\star = \boldsymbol{S}_\star + \boldsymbol{L}_\star$. To estimate $\boldsymbol{S}_\star$ and $\boldsymbol{L}_\star$, we solve the optimization problem (4) via AltPGD (summarized in Algorithm 2).

We set $f_n(\boldsymbol{S}, \boldsymbol{L}) = \|\boldsymbol{Y} - \boldsymbol{X}(\boldsymbol{S} + \boldsymbol{L})\|_F^2/(2n)$ and write the update of AltPGD as

$$\begin{aligned} \boldsymbol{S}_{k+1} &= \mathcal{P}_{\mathcal{K}_S}(\boldsymbol{S}_k - \mu\nabla_S f_n(\boldsymbol{S}_k, \boldsymbol{L}_k)), \\ \boldsymbol{L}_{k+1} &= \mathcal{P}_{\mathcal{K}_L}(\boldsymbol{L}_k - \mu\nabla_L f_n(\boldsymbol{S}_k, \boldsymbol{L}_k)), \end{aligned} \tag{22}$$

where $\mathcal{K}_S = \{\boldsymbol{S} \mid \mathcal{R}_S(\boldsymbol{S}) \leq \mathcal{R}_S(\boldsymbol{S}_\star)\}$ and $\mathcal{K}_L = \{\boldsymbol{L} \mid \mathcal{R}_L(\boldsymbol{L}) \leq \mathcal{R}_L(\boldsymbol{L}_\star)\}$. We also introduce two descent cones $\mathcal{C}_S = \text{cone}(\mathcal{K}_S - \boldsymbol{S}_\star)$ and $\mathcal{C}_L = \text{cone}(\mathcal{K}_L - \boldsymbol{L}_\star)$, which would be used in our analysis.

---

**Algorithm 2** AltPGD for superposition-structured transition matrices estimation
***
   **Input:** Initial points $\boldsymbol{S}_0$ and $\boldsymbol{L}_0$, step size $\mu$, iteration number $K$.
   **for** $k = 0$ **to** $K - 1$ **do**
      $\boldsymbol{S}_{k+1} = \mathcal{P}_{\mathcal{K}_S}(\boldsymbol{S}_k - \mu\nabla_S f_n(\boldsymbol{S}_k, \boldsymbol{L}_k))$
      $\boldsymbol{L}_{k+1} = \mathcal{P}_{\mathcal{K}_L}(\boldsymbol{L}_k - \mu\nabla_L f_n(\boldsymbol{S}_k, \boldsymbol{L}_k))$
   **end for**
   **Output:** $\boldsymbol{S}_K$ and $\boldsymbol{L}_K$

---

In this part, we consider $\mathcal{R}_S(\cdot)$ and $\mathcal{R}_L(\cdot)$ both belong to decomposable norms defined in [14].

---

[1]In addition, our analysis framework could also be used in the problem to estimate $\boldsymbol{\Gamma}_\star$ and the precision matrix $\boldsymbol{\Sigma}_e^{-1}$ simultaneously [28].

**Definition 1** (Decomposable norm). *A regularization function $\mathcal{R}(\cdot)$ is decomposable with respect to a subspace pair $(\mathcal{M}, \overline{\mathcal{M}}^{\perp})$, if*

$$\mathcal{R}(\alpha + \beta) = \mathcal{R}(\alpha) + \mathcal{R}(\beta), \quad \forall \alpha \in \mathcal{M}, \ \beta \in \overline{\mathcal{M}}^{\perp}. \tag{23}$$

*Here, $\mathcal{M}$ is referred to as the model subspace which captures the constraints determined by the model and $\mathcal{M} \subseteq \overline{\mathcal{M}}$. $\overline{\mathcal{M}}^{\perp}$ is called the perturbation subspace indicating the deviation from the model subspace $\mathcal{M}$.*

For common structure priors such as sparsity, group-sparsity and low-rank property, the corresponding regularization functions $l_1$-norm, $l_{1,2}$-norm and nuclear norm all belong to decomposable norms with low-dimensional model subspaces.

For the superposition-structured transition matrix $\boldsymbol{\Gamma}_{\star} = \boldsymbol{S}_{\star} + \boldsymbol{L}_{\star}$, we assume $\mathcal{R}_S(\cdot)$ is decomposable with respect to a subspace pair $(\mathcal{M}_{\mathcal{S}}, \overline{\mathcal{M}}_{\mathcal{S}}^{\perp})$, which is suit for the single-structured parameter $\boldsymbol{S}_{\star}$. Similarly, $\mathcal{R}_L(\cdot)$ is decomposable with respect to a subspace pair $(\mathcal{M}_{\mathcal{L}}, \overline{\mathcal{M}}_{\mathcal{L}}^{\perp})$ suit for $\boldsymbol{L}_{\star}$.

Due to the superposition-structured property of the transition matrix, we need to impose an additional assumption about the interaction between the two different structured components to guarantee the separate estimation.

**Assumption 3** (Structural incoherence). *Given the subspace pairs $(\mathcal{M}_{\mathcal{S}}, \overline{\mathcal{M}}_{\mathcal{S}}^{\perp})$ and $(\mathcal{M}_{\mathcal{L}}, \overline{\mathcal{M}}_{\mathcal{L}}^{\perp})$ for the two parameters $\boldsymbol{S}_{\star}$ and $\boldsymbol{L}_{\star}$. Suppose the covariance matrix $\boldsymbol{\Sigma}_x$ defined in (9) satisfies*

$$\max \left\{ \bar{\sigma}_{\max}(\mathcal{P}_{\overline{\mathcal{M}}_S} \boldsymbol{\Sigma}_x \mathcal{P}_{\overline{\mathcal{M}}_L}), \bar{\sigma}_{\max}(\mathcal{P}_{\overline{\mathcal{M}}_S^{\perp}} \boldsymbol{\Sigma}_x \mathcal{P}_{\overline{\mathcal{M}}_L}), \right.$$
$$\left. \bar{\sigma}_{\max}(\mathcal{P}_{\overline{\mathcal{M}}_S} \boldsymbol{\Sigma}_x \mathcal{P}_{\overline{\mathcal{M}}_L^{\perp}}), \bar{\sigma}_{\max}(\mathcal{P}_{\overline{\mathcal{M}}_S^{\perp}} \boldsymbol{\Sigma}_x \mathcal{P}_{\overline{\mathcal{M}}_L^{\perp}}) \right\} \leq \frac{\kappa_{\min}}{8}, \tag{24}$$

*where $\bar{\sigma}_{\max}(\cdot)$ for a matrix $\boldsymbol{\Sigma}$ is defined as $\bar{\sigma}_{\max}(\boldsymbol{\Sigma}) = \sup\limits_{\boldsymbol{V}, \boldsymbol{U} \in \mathbb{S}_F} \langle \boldsymbol{V}, \boldsymbol{\Sigma U} \rangle$ and $\kappa_{\min}$ is defined in (12). Here $\mathcal{P}_{\overline{\mathcal{M}}_S}$, $\mathcal{P}_{\overline{\mathcal{M}}_L}$, $\mathcal{P}_{\overline{\mathcal{M}}_S^{\perp}}$ and $\mathcal{P}_{\overline{\mathcal{M}}_L^{\perp}}$ donate the orthogonal projection operators onto the corresponding subspaces.*

**Remark 5** (Related works). *Several similar assumptions have been imposed in [27, 29, 30]. This type of assumptions is first proposed by Yang and Ravikumar in [27], where they use the structural incoherence assumption to restrict the interaction between different components of superposition-structured statistical models and the C-Linear condition guarantees the structural incoherence under the linear regression setting and the Gaussian design. Meng et al. generalize the C-Linear assumption to the Structural Fisher Incoherence assumption in [29] for the estimation of sparse plus low-rank matrices in Gaussian Graphical Models. In [30], Greenewald and Hero introduce the structural incoherence assumption to robust Kronecker product PCA models. Our Assumption 3 is also motivated by [27] and would reduce to the C-Linear condition in [27] when we consider $\Sigma$-Gaussian ensemble where the rows of $\boldsymbol{X}$ in (2) are generated independently from $\mathcal{N}(\boldsymbol{0}, \boldsymbol{\Sigma}_x)$.*

**Remark 6** (Nonidentifiability). *There are also other conditions used in literature to deal with the nonidentifiability concern. In [22], Basu et al. refer to the spikiness condition, which is first introduced in [31] for matrix completion and then is extended to matrix decomposition in [32]. In [27], Yang and Ravikumar compare the structure incoherence used here with the spikiness condition and illustrate the structure incoherence could address a drawback of the spikiness condition that the estimation error does not approach zero when the number of samples approaches infinity and requires weaker conditions at the same time. Another common condition used in [23–26] for sparse plus low-rank matrices recovery is the incoherence condition which is first proposed in [33, 34]. In [31, 32], the authors illustrate that the spikiness condition is a milder condition than the incoherence condition and is more suitable for the noisy models because of the consideration of singular values.*

We now present the non-asymptotic optimization guarantee of AltPGD for the problem (4).

**Theorem 2.** *Consider the VAR model (1) satisfying Assumptions 1,2 and 3. Suppose $\boldsymbol{\Gamma}_{\star}$ is superposition-structured and $\boldsymbol{\Gamma}_{\star} = \boldsymbol{S}_{\star} + \boldsymbol{L}_{\star}$, where $\boldsymbol{S}_{\star}$ and $\boldsymbol{L}_{\star}$ are two single-structured parameters whose structures are characterized by two decomposable norms $\mathcal{R}_S(\cdot)$ and $\mathcal{R}_L(\cdot)$ respectively. Starting from points $\boldsymbol{S}_0$ and $\boldsymbol{L}_0$ satisfying $\mathcal{R}_S(\boldsymbol{S}_0) \leq \mathcal{R}_S(\boldsymbol{S}_{\star})$ and $\mathcal{R}_L(\boldsymbol{L}_0) \leq \mathcal{R}_L(\boldsymbol{L}_{\star})$, we solve the optimization problem (4) via AltPGD with the step size $\mu = 1/\kappa_{\max}$. If the number of measurements satisfies*

$$\sqrt{n} > 4C \frac{\kappa_{\max}}{\kappa_{\min}} (\omega(\mathcal{C}_S \cap \mathbb{S}_F) + \omega(\mathcal{C}_L \cap \mathbb{S}_F) + u), \tag{25}$$

*then the update* (22) *would obey*

$$\|\boldsymbol{S}_{k+1} - \boldsymbol{S}_\star\|_{\mathrm{F}} + \|\boldsymbol{L}_{k+1} - \boldsymbol{L}_\star\|_{\mathrm{F}} \leq \rho^{k+1}(\|\boldsymbol{S}_0 - \boldsymbol{S}_\star\|_{\mathrm{F}} + \|\boldsymbol{L}_0 - \boldsymbol{L}_\star\|_{\mathrm{F}}) + \frac{\xi}{1-\rho} \qquad (26)$$

*with probability at least* $1 - c\exp(-u^2)$. *Here*

$$\rho = 1 - \frac{3}{4}\frac{\kappa_{\min}}{\kappa_{\max}} + C\frac{\omega(\mathcal{C}_L \cap \mathbb{S}_F) + \omega(\mathcal{C}_S \cap \mathbb{S}_F) + u}{\sqrt{n}} < 1 - \frac{\kappa_{\min}}{2\kappa_{\max}}, \qquad (27)$$

$$\frac{\xi}{1-\rho} = \frac{1}{1-\rho} \cdot C'\frac{1}{\sqrt{\kappa_{\max}}}\|\boldsymbol{\Sigma}_e\|^{\frac{1}{2}}\frac{\omega(\mathcal{C}_L \cap \mathbb{S}_F) + \omega(\mathcal{C}_S \cap \mathbb{S}_F) + u}{\sqrt{n}}$$

$$< 2C'\frac{\sqrt{\kappa_{\max}}}{\kappa_{\min}}\|\boldsymbol{\Sigma}_e\|^{\frac{1}{2}}\frac{\omega(\mathcal{C}_L \cap \mathbb{S}_F) + \omega(\mathcal{C}_S \cap \mathbb{S}_F) + u}{\sqrt{n}}, \qquad (28)$$

*and* $c$, $C$, $C'$ *are absolute constants.*

**Remark 7** (Related works). *Our analysis makes progress on three aspects compared with the result in [22]. First, we illustrate the linear convergence rate of AltPGD compared with the sub-linear rate of FNSL. Second, our analysis indicates the requirement of samples for the linear rate and the statistical error, which are absent in the analysis of optimization in [22]. Third, Theorem 2 addresses a drawback of the result in [22] that the estimation error does not converge to zero when the number of samples approaches infinity.*

Our analysis framework is also valid for robust PCA considered in [32, 27]. Suppose we have $n$ i.i.d. sample $\boldsymbol{z}_i \in \mathbb{R}^d$, where $\boldsymbol{z}_i = \boldsymbol{u}_i + \boldsymbol{v}_i$, $\boldsymbol{u}_i \sim \mathcal{N}(\boldsymbol{0}, \boldsymbol{L}_\star)$, $\boldsymbol{v}_i \sim \mathcal{N}(\boldsymbol{0}, \boldsymbol{S}_\star)$, $\boldsymbol{u}_i$ and $\boldsymbol{v}_i$ are independent. Here we set $\boldsymbol{L}_\star$ is a low-rank matrix and $\boldsymbol{S}_\star$ is a sparse matrix. We could write the sample matrix as $\boldsymbol{Y} = \frac{1}{n}\sum_{i=1}^n \boldsymbol{z}_i\boldsymbol{z}_i^T = \boldsymbol{L}_\star + \boldsymbol{S}_\star + \boldsymbol{E}$, where $\boldsymbol{E} = \frac{1}{n}\sum_{i=1}^n \boldsymbol{z}_i\boldsymbol{z}_i^T - (\boldsymbol{L}_\star + \boldsymbol{S}_\star)$ is a Wishart noise matrix. In this setting, we have the data matrix $\boldsymbol{X} = \boldsymbol{I}_{d\times d}$ and solve the problem

$$\min_{\boldsymbol{S},\boldsymbol{L}} \quad \frac{1}{2}\|\boldsymbol{Y} - \boldsymbol{S} - \boldsymbol{L}\|_{\mathrm{F}}^2,$$
$$\text{s.t.} \quad \|\mathrm{vec}(\boldsymbol{S}^T)\|_1 \leq \|\mathrm{vec}(\boldsymbol{S}_\star^T)\|_1, \quad \|\boldsymbol{L}\|_\star \leq \|\boldsymbol{L}_\star\|_\star, \qquad (29)$$

where $\|\cdot\|_1$ represents the $l_1$-norm of a vector and $\|\cdot\|_\star$ represents the nuclear norm of a matrix.

**Corollary 2.** *Consider the robust PCA model where $\boldsymbol{S}_\star$ is a sparse matrix with $s_\star$ non-zero entries and $\boldsymbol{L}_\star$ is a $r_\star$-rank matrix. Under Assumption 3 where $\boldsymbol{\Sigma}_x = \boldsymbol{I}_{d\times d}$ and $\kappa_{\min} = \kappa_{\max} = 1$ in this setting, we solve the optimization problem* (29) *via AltPGD with the step size $\mu = 1$ and starting points $\boldsymbol{S}_0$ and $\boldsymbol{L}_0$ satisfying $\|\mathrm{vec}(\boldsymbol{S}_0^T)\|_1 \leq \|\mathrm{vec}(\boldsymbol{S}_\star^T)\|_1$ and $\|\boldsymbol{L}_0\|_\star \leq \|\boldsymbol{L}_\star\|_\star$. If the number of measurements satisfies*

$$\sqrt{n} > C'(\sqrt{s_\star \log d} + \sqrt{r_\star d} + u), \qquad (30)$$

*then the update of AltPGD would obey*

$$\|\boldsymbol{S}_{k+1} - \boldsymbol{S}_\star\|_{\mathrm{F}} + \|\boldsymbol{L}_{k+1} - \boldsymbol{L}_\star\|_{\mathrm{F}}$$
$$\leq (\frac{1}{4})^{k+1}(\|\boldsymbol{S}_0 - \boldsymbol{S}_\star\|_{\mathrm{F}} + \|\boldsymbol{L}_0 - \boldsymbol{L}_\star\|_{\mathrm{F}}) + \frac{4}{3}C\|\boldsymbol{S}_\star + \boldsymbol{L}_\star\|\frac{\sqrt{s_\star \log d} + \sqrt{r_\star d} + u}{\sqrt{n}}, \qquad (31)$$

*with probability at least* $1 - c\exp(-u^2)$. *Here $c$, $C$ and $C'$ are absolute constants.*

## 5 Numerical results

### 5.1 Synthetic data

In this section, we apply PGD and AltPGD[1] to network learning problems and compare the performance with FNSL proposed in [22]. We regularize sparse matrices by the $l_1$-norm and low-rank matrices by the nuclear norm.[2] All simulations are run on a PC with Intel i5-6500 and 16GB memory.

---

[1]The step size is selected as the inverse of the maximum eigenvalue of the estimated covariance matrix of the samples and the estimated covariance matrix is derived from $\boldsymbol{X}^T\boldsymbol{X}/n$.

[2]The projection onto the $l_1$-norm ball follows the procedure in [35] and the projection onto the nuclear norm ball is a union of the singular value decomposition and the projection onto the $l_1$-norm ball.

We first introduce several performance metrics for network estimation. For the estimated transition matrix $\hat{\mathbf{\Gamma}}$ and the real transition matrix $\mathbf{\Gamma}_\star$ whose entries are denoted by $\hat{\gamma}_{ij}$ and $\gamma_{ij}^\star$ respectively, we define the true positive rate (TPR) and false alarm rate (FAR) as

$$\text{TPR} \coloneqq \frac{\sharp\{\hat{\gamma}_{ij} \neq 0 \text{ and } \gamma_{ij}^\star \neq 0\}}{\sharp\{\gamma_{ij}^\star \neq 0\}}, \qquad \text{FAR} \coloneqq \frac{\sharp\{\hat{\gamma}_{ij} \neq 0 \text{ and } \gamma_{ij}^\star = 0\}}{\sharp\{\gamma_{ij}^\star = 0\}}.$$

We also introduce the estimation error (EE), where $\text{EE} \coloneqq \|\hat{\mathbf{\Gamma}} - \mathbf{\Gamma}_\star\|_{\text{F}} / \|\mathbf{\Gamma}_\star\|_{\text{F}}$.

### 5.1.1 Network learning with a sparse transition matrix

First we consider $\mathbf{\Gamma}_\star \in \mathbb{R}^{d \times d}$ is a sparse matrix with $s_\star$ non-zero entries. We suppose each row of $\mathbf{\Gamma}_\star$ has $s_\star/d$ non-zero entries whose values follow a standard normal distribution. Then we rescale $\mathbf{\Gamma}_\star$ to guarantee the stability of the process. In this simulation, we set $d = 100$ and $s_\star = 3500$. To illustrate the effect of the numbers of samples, we perform the simulation under three scenarios $n = 1000, 1500, 2000$ and each scenario is repeated for 100 trials. For FNSL, we choose the regularization parameter $\lambda_S$ as $\mathcal{O}(\sqrt{n \log d})$ according to Propositions 1 and 3 in [22]. Both algorithms start from $\mathbf{\Gamma}_0 = \mathbf{0}$.

Table 1: Performance comparison between PGD and FNSL on sparse network learning problems

| $d = 100$ | Method | TPR (%) | FAR (%) | EE | Total time $(s)$ |
|---|---|---|---|---|---|
| $n = 1000$ | PGD | **79.49** | **11.04** | **0.476** | **3.18** |
| | FNSL | 73.64 | 14.19 | 0.489 | 75.59 |
| $n = 1500$ | PGD | **83.45** | **8.91** | **0.396** | **5.16** |
| | FNSL | 78.43 | 11.62 | 0.417 | 140.16 |
| $n = 2000$ | PGD | **85.82** | **7.63** | **0.350** | **6.14** |
| | FNSL | 81.30 | 10.07 | 0.373 | 183.79 |

In Table 1, we record the experimental results for the two algorithms under different numbers of samples. The results illustrate that PGD enjoys better performance with much less computation time than FNSL and support our analysis in Theorem 1.

### 5.1.2 Network learning with a low-rank transition matrix

Then we consider $\mathbf{\Gamma}_\star \in \mathbb{R}^{d \times d}$ is a low-rank matrix whose rank is $r_\star$. Suppose $\mathbf{\Gamma}_\star$ is constructed by $\mathbf{\Gamma}_\star = \mathbf{U} \mathbf{V}^T$, where $\mathbf{U}, \mathbf{V} \in \mathbb{R}^{d \times r_\star}$ are matrices with independent standard Gaussian entries. We also rescale $\mathbf{\Gamma}_\star$ to guarantee the stability of the process. We set $d = 100$, $n = 8000$, $r_\star = 2$ and repeat the scenario for 100 times. For FNSL, we choose the regularization parameter $\lambda_L$ as $\mathcal{O}(\sqrt{nd})$ according to Propositions 1 and 3 in [22]. The result in Figure 1(a) illustrates that the updates of PGD enjoy a faster convergence rate than those of FNSL, as predicted in Theorem 1.

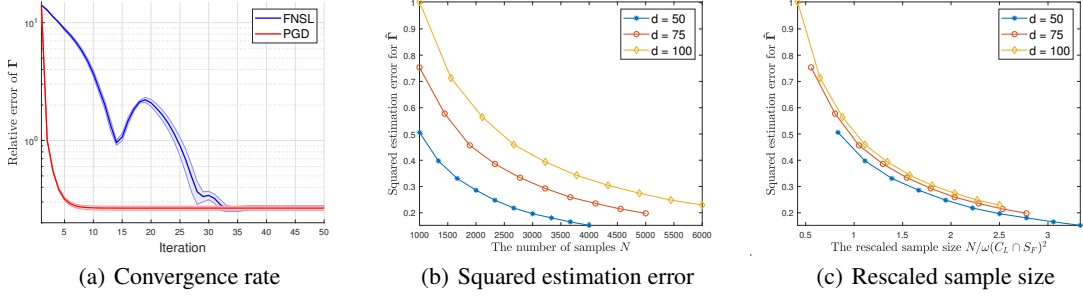

| (a) Convergence rate | (b) Squared estimation error | (c) Rescaled sample size |
|---|---|---|

Figure 1: Convergence results of PGD for low-rank transition matrices estimation.

We also perform simulations under different dimensions $d$ and different numbers of samples $n$ to verify the order of estimation error, where we set $r_\star = 4$ and each scenario is repeated for 100 times. The results in Figure 1(b) and 1(c) indicate that all the squared empirical error curves behave as $f(t) \propto t^{-1}$ and support our theoretical results in Theorem 1.

### 5.1.3 Network learning with a superposition-structured transition matrix

In this part, we suppose the transition matrix in the VAR model (1) is superposition-structured. Specially, we consider $\boldsymbol{\Gamma}_\star = \boldsymbol{S}_\star + \boldsymbol{L}_\star$, where $\boldsymbol{S}_\star$ is a sparse matrix with $s_\star$ non-zero entries and $\boldsymbol{L}_\star$ is a rank-$r_\star$ matrix. The construction of $\boldsymbol{S}_\star$ and $\boldsymbol{L}_\star$ follows the same procedure as the above simulations. First, we set $d = 100$, $s_\star = 3500$ and $r_\star = 2$. To illustrate the effect of the numbers of samples, we perform the simulation under three scenarios $n = 1500, 2000, 2500$ and each scenario is repeated for 100 trials. For FNSL, we choose the regularization parameters $\lambda_S$ as $\mathcal{O}(\sqrt{n \log d})$ and $\lambda_L$ as $\mathcal{O}(\sqrt{nd})$ according to Propositions 1 and 3 in [22]. Both algorithms start from $\boldsymbol{S}_0 = \boldsymbol{0}$ and $\boldsymbol{L}_0 = \boldsymbol{0}$.

Table 2: Performance comparison between AltPGD and FNSL on estimation of sparse plus low-rank transition matrices

| $d = 100$ | Method | TPR (%) | FAR (%) | EE | Total time ($s$) |
|---|---|---|---|---|---|
| $n = 1500$ | AltPGD | **78.26** | **11.70** | **0.475** | **19.16** |
| | FNSL | 71.18 | 15.52 | 0.486 | 309.76 |
| $n = 2000$ | AltPGD | **81.06** | **10.20** | **0.421** | **26.05** |
| | FNSL | 74.65 | 13.65 | 0.438 | 436.46 |
| $n = 2500$ | AltPGD | **83.19** | **9.05** | **0.379** | **32.27** |
| | FNSL | 77.49 | 12.12 | 0.399 | 544.08 |

In Table 2, we record the experimental results for the two algorithms under different numbers of samples. The results illustrate that AltPGD enjoys better performance with much less computation time than FNSL.

Then we compare the convergence rates of AltPGD and FNSL. We set $d = 100$, $n = 8000$, $s_\star = 3500$, $r_\star = 3$ and repeat the scenario for 100 times. The result in Figure 2(a) illustrates the efficiency of AltPGD.

We also perform simulations under different dimensions $d$ and different numbers of samples $n$ to verify the order of estimation error, where we set $s_\star = 300$, $r_\star = 3$ and each scenario is repeated for 100 times. The results in Figure 2(b) and 2(c) indicate that all the squared empirical error curves behave as $f(t) \propto t^{-1}$ and support our theoretical results in Theorem 2.

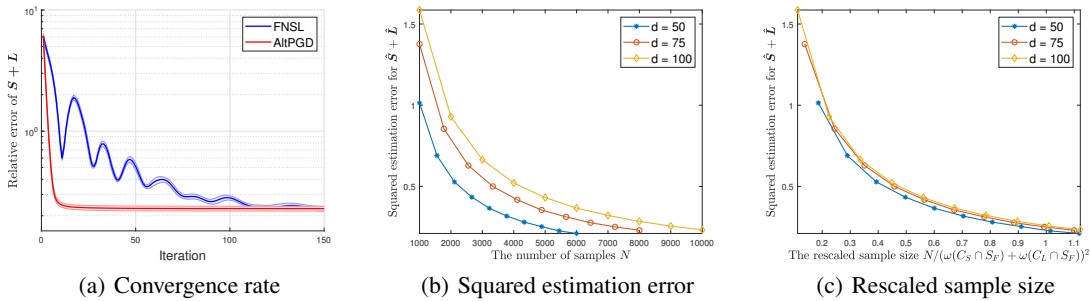

(a) Convergence rate      (b) Squared estimation error      (c) Rescaled sample size

Figure 2: Convergence results of AltPGD for sparse plus low-rank transition matrices estimation.

### 5.2 Real data

Next, we analyze the temporal dynamics of the log-returns of stocks in the S&P 500 index. The stock data consists of 1259 daily closing prices for 434 companies in the S&P 500 index between February 8, 2013 and February 7, 2018 [36]. In this way, we get 1259 data vectors, each of which contains the closing prices of all stocks on a trading day. To ensure the data stationary, we calculate the log-returns $\{\boldsymbol{r}_t\}_{t=1}^{T-1}$ of stocks by

$$r_{t,i} = \log(\frac{p_{t+1,i}}{p_{t,i}}), \quad t = 1, \cdots, T-1, \tag{32}$$

where $p_{t,i}$ is the closing price of stock $i$ at day $t$. In this way, we construct the data matrix $\boldsymbol{X} = [\boldsymbol{r}_1, \cdots, \boldsymbol{r}_{T-2}]^T$ and the observation matrix $\boldsymbol{Y} = [\boldsymbol{r}_2, \cdots, \boldsymbol{r}_{T-1}]^T$.

We adopt the VAR model (1) with the regularizer $\mathcal{R}(\cdot) = \|\cdot\|_1$ to study the evolution of stock log-returns over the 2013-2018 period and then solve the model with PGD. The constrained parameter is selected through 5-fold cross validation. In Figure 3, we present the sparsity patterns of two parts of the transition matrix $\hat{\boldsymbol{\Gamma}}$ estimated by PGD, which indicate meaningful Granger causal effects [37, 38] among the log-returns of stocks. The 434 companies belong to 10 different sectors, such as materials (22 stocks), energy (29 stocks), consumer staples (31 stocks) and financials (83 stocks). In Figure 3(a) and 3(b), the log-returns of stocks in the energy sector have stronger Granger causal effects, and the Granger causal effects of the consumer staples sector and the financials sector are weaker. For comparison, we present the sparsity patterns of the transition matrix estimated by FNSL in [22], which illustrate a similar Granger causal network.

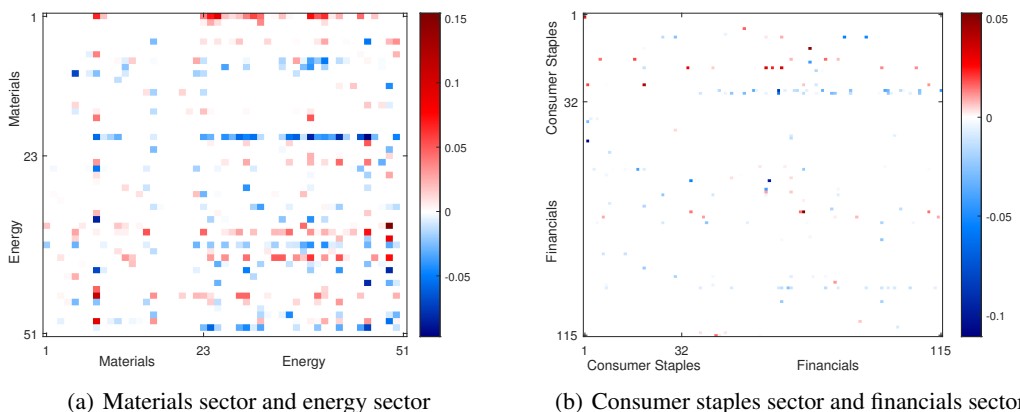

(a) Materials sector and energy sector         (b) Consumer staples sector and financials sector

Figure 3: Sparsity patterns of the transition matrix $\hat{\boldsymbol{\Gamma}}$ estimated by PGD.

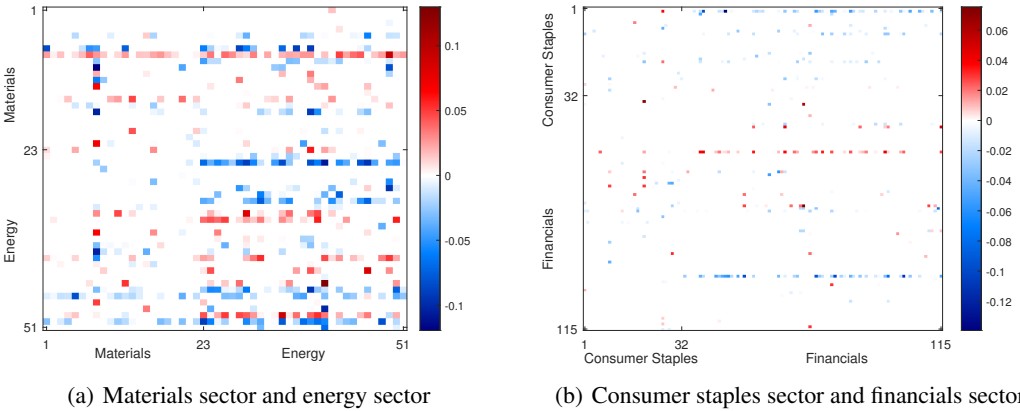

(a) Materials sector and energy sector         (b) Consumer staples sector and financials sector

Figure 4: Sparsity patterns of the transition matrix $\hat{\boldsymbol{\Gamma}}$ estimated by FNSL.

## Acknowledgments and Disclosure of Funding

The authors would like to thank the anonymous reviewers for their constructive feedback and suggestions. This work was supported by the National Natural Science Foundation of China under Grants 61971044 and 62025103.

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
