# Supplementary Material to Linear Convergence of Gradient Methods for Estimating Structured Transition Matrices in High-dimensional Vector Autoregressive Models

We use $C$, $C'$, $C''$, $C'''$ and $c$ to denote positive constants which might change from line to line throughout this part.

## A Proof of main theorems

### A.1 Proof of Theorem 1

First we provide several expression used in our analysis.

$$f_n(\boldsymbol{\Gamma}) = \frac{1}{2n}\|\boldsymbol{Y} - \boldsymbol{X}\boldsymbol{\Gamma}\|_{\mathrm{F}}^2, \tag{33}$$

$$f(\boldsymbol{\Gamma}) = \frac{1}{2}\operatorname{tr}\left\{(\boldsymbol{\Gamma} - \boldsymbol{\Gamma}_\star)^T\boldsymbol{\Sigma}_x(\boldsymbol{\Gamma} - \boldsymbol{\Gamma}_\star) + \boldsymbol{I}\right\}, \tag{34}$$

$$\nabla f_n(\boldsymbol{\Gamma}) = \frac{1}{n}\boldsymbol{X}^T\boldsymbol{X}(\boldsymbol{\Gamma} - \boldsymbol{\Gamma}_\star) - \frac{1}{n}\boldsymbol{X}^T\boldsymbol{E}, \tag{35}$$

$$\nabla f(\boldsymbol{\Gamma}) = \boldsymbol{\Sigma}_x(\boldsymbol{\Gamma} - \boldsymbol{\Gamma}_\star). \tag{36}$$

**Lemma 2.** *Suppose $\bar{\boldsymbol{x}} = \mathcal{P}_\mathcal{K}(\boldsymbol{y})$, where $\mathcal{K} = \{\boldsymbol{x} \mid \mathcal{R}(\boldsymbol{x}) \le \mathcal{R}(\boldsymbol{x}^\star)\}$ and $\mathcal{R}(\cdot)$ is a convex function. Then we could bound $\|\bar{\boldsymbol{x}} - \boldsymbol{x}^\star\|_2$ as*

$$\|\bar{\boldsymbol{x}} - \boldsymbol{x}^\star\|_2 \le \sup_{\boldsymbol{v} \in \mathcal{C} \cap \mathbb{S}_2} \langle \boldsymbol{v}, \boldsymbol{y} - \boldsymbol{x}^\star\rangle, \tag{37}$$

*where $\mathcal{C} = \operatorname{cone}(\mathcal{D})$ is the decent cone, $\mathcal{D} = \mathcal{K} - \{\boldsymbol{x}^\star\}$ is the descent set and $\mathbb{S}_2$ is the sphere with unit Euclidean norm.*

In this way, we could bound the difference between $\boldsymbol{\Gamma}_{k+1}$ and $\boldsymbol{\Gamma}_\star$ as

$$
\begin{aligned}
&\|\boldsymbol{\Gamma}_{k+1} - \boldsymbol{\Gamma}_\star\|_{\mathrm{F}} \\
&= \|\mathcal{P}_\mathcal{K}(\boldsymbol{\Gamma}_k - \mu\nabla f_n(\boldsymbol{\Gamma}_k)) - \boldsymbol{\Gamma}_\star\|_{\mathrm{F}} \\
&\le \sup_{\boldsymbol{V} \in \mathcal{C} \cap \mathbb{S}_F} \langle \boldsymbol{V}, \boldsymbol{\Gamma}_k - \boldsymbol{\Gamma}_\star - \mu\nabla f_n(\boldsymbol{\Gamma}_k)\rangle \\
&= \sup_{\boldsymbol{V} \in \mathcal{C} \cap \mathbb{S}_F} \langle \boldsymbol{V}, \boldsymbol{\Gamma}_k - \boldsymbol{\Gamma}_\star - \mu\frac{1}{n}\boldsymbol{X}^T\boldsymbol{X}(\boldsymbol{\Gamma}_k - \boldsymbol{\Gamma}_\star) + \mu\frac{1}{n}\boldsymbol{X}^T\boldsymbol{E}\rangle \\
&= \sup_{\boldsymbol{V} \in \mathcal{C} \cap \mathbb{S}_F} \langle \boldsymbol{V}, \boldsymbol{\Gamma}_k - \boldsymbol{\Gamma}_\star - \mu\boldsymbol{\Sigma}_x(\boldsymbol{\Gamma}_k - \boldsymbol{\Gamma}_\star) + \mu(\boldsymbol{\Sigma}_x - \frac{1}{n}\boldsymbol{X}^T\boldsymbol{X})(\boldsymbol{\Gamma}_k - \boldsymbol{\Gamma}_\star) + \mu\frac{1}{n}\boldsymbol{X}^T\boldsymbol{E}\rangle \\
&\le \underbrace{\sup_{\boldsymbol{V} \in \mathcal{C} \cap \mathbb{S}_F} \langle \boldsymbol{V}, (\boldsymbol{I} - \mu\boldsymbol{\Sigma}_x)(\boldsymbol{\Gamma}_k - \boldsymbol{\Gamma}_\star)\rangle}_{I} + \underbrace{\sup_{\boldsymbol{V} \in \mathcal{C} \cap \mathbb{S}_F} \langle \boldsymbol{V}, \mu(\boldsymbol{\Sigma}_x - \frac{1}{n}\boldsymbol{X}^T\boldsymbol{X})(\boldsymbol{\Gamma}_k - \boldsymbol{\Gamma}_\star)\rangle}_{II} \\
&\quad + \underbrace{\sup_{\boldsymbol{V} \in \mathcal{C} \cap \mathbb{S}_F} \langle \boldsymbol{V}, \mu\frac{1}{n}\boldsymbol{X}^T\boldsymbol{E}\rangle}_{III},
\end{aligned}
\tag{38}
$$

where the first inequality refers to Lemma 2 and $\mathcal{C} = \operatorname{cone}(\mathcal{K} - \{\boldsymbol{\Gamma}_\star\})$ is the descent cone generated from $\mathcal{K} = \{\boldsymbol{\Gamma} \mid \mathcal{R}(\boldsymbol{\Gamma}) \le \mathcal{R}(\boldsymbol{\Gamma}_\star)\}$.

The first term I of (38) only relates to the population loss function $f(\boldsymbol{\Gamma}_k)$, which could be bounded according to the following lemma.

**Lemma 3.** *Under Assumption 2, the term $\boldsymbol{\Gamma} - \boldsymbol{\Gamma}_\star - \mu\nabla f(\boldsymbol{\Gamma})$ about the gradient of the population loss function (34) with the step size $\mu = 1/\kappa_{\max}$ satisfies*

$$\|\boldsymbol{\Gamma} - \boldsymbol{\Gamma}_\star - \mu\nabla f(\boldsymbol{\Gamma})\|_{\mathrm{F}} \le (1 - \frac{\kappa_{\min}}{\kappa_{\max}})\|\boldsymbol{\Gamma} - \boldsymbol{\Gamma}_\star\|_{\mathrm{F}}. \tag{39}$$

Then we could bound the first term I of (38) as

$$\sup_{\boldsymbol{V} \in \mathcal{C} \cap \mathbb{S}_F} \langle \boldsymbol{V}, (\boldsymbol{I} - \mu\boldsymbol{\Sigma}_x)(\boldsymbol{\Gamma}_k - \boldsymbol{\Gamma}_\star)\rangle \leq \|(\boldsymbol{I} - \mu\boldsymbol{\Sigma}_x)(\boldsymbol{\Gamma}_k - \boldsymbol{\Gamma}_\star)\|_{\mathrm{F}}$$

$$= \|\boldsymbol{\Gamma}_k - \boldsymbol{\Gamma}_\star - \mu\nabla f(\boldsymbol{\Gamma}_k)\|_{\mathrm{F}}$$

$$\leq (1 - \frac{\kappa_{\min}}{\kappa_{\max}})\|\boldsymbol{\Gamma}_k - \boldsymbol{\Gamma}_\star\|_{\mathrm{F}}, \tag{40}$$

where the first inequality is from the Cauchy-Schwarz inequality and the last inequality is based on Lemma 3.

The second part II of (38) is about the deviation inequality of the sample Gram matrix $\boldsymbol{X}^T\boldsymbol{X}/n$. The following lemma would help us deal with the supremum of the quadratic form.

**Lemma 4.** *Suppose* $vec(\boldsymbol{X}^T)$ *follows the distribution* $\mathcal{N}(\boldsymbol{0}, \boldsymbol{\Upsilon}_x)$ *where* $\lambda_{\max}(\boldsymbol{\Upsilon}_x) \leq \kappa_{\max}$ *and the rows of* $\boldsymbol{X}$ *follow the distribution* $\mathcal{N}(\boldsymbol{0}, \boldsymbol{\Sigma}_x)$. *Under the condition of* $n \geq (\omega(\mathcal{C}_1 \cap \mathbb{S}_F) + \omega(\mathcal{C}_2 \cap \mathbb{S}_F) + u)^2$, *we have*[1]

$$P\Big(\sup_{\substack{\boldsymbol{U} \in \mathcal{C}_1 \cap \mathbb{S}_F \\ \boldsymbol{V} \in \mathcal{C}_2 \cap \mathbb{S}_F}} |\langle \boldsymbol{V}, (\boldsymbol{\Sigma}_x - \frac{\boldsymbol{X}^T\boldsymbol{X}}{n})\boldsymbol{U}\rangle| > C\kappa_{\max} \frac{\omega(\mathcal{C}_1 \cap \mathbb{S}_F) + \omega(\mathcal{C}_2 \cap \mathbb{S}_F) + u}{\sqrt{n}}\Big) \leq 2\exp(-u^2),$$

$$\tag{41}$$

*where* $C$ *is a constant,* $\mathcal{C}_1$ *and* $\mathcal{C}_2$ *are two convex cones.*

The condition $\lambda_{\max}(\boldsymbol{\Upsilon}_x) \leq \kappa_{\max}$ is satisfied according to Assumption 2 and Lemma 1. By setting $\mathcal{C}_1 = \mathcal{C}_2 = \mathcal{C}$, we could bound the second part II of (38) with Lemma 4 under the condition of $n \geq (2\omega(\mathcal{C} \cap \mathbb{S}_F) + u)^2$ as

$$\sup_{\boldsymbol{V} \in \mathcal{C} \cap \mathbb{S}_F} \langle \boldsymbol{V}, \mu(\boldsymbol{\Sigma}_x - \frac{1}{n}\boldsymbol{X}^T\boldsymbol{X})(\boldsymbol{\Gamma}_k - \boldsymbol{\Gamma}_\star)\rangle \leq \sup_{\boldsymbol{U}, \boldsymbol{V} \in \mathcal{C} \cap \mathbb{S}_F} \langle \boldsymbol{V}, \mu(\boldsymbol{\Sigma}_x - \frac{1}{n}\boldsymbol{X}^T\boldsymbol{X})\boldsymbol{U}\rangle\|\boldsymbol{\Gamma}_k - \boldsymbol{\Gamma}_\star\|_{\mathrm{F}}$$

$$\leq \mu C\kappa_{\max} \frac{2\omega(\mathcal{C} \cap \mathbb{S}_F) + u}{\sqrt{n}}\|\boldsymbol{\Gamma}_k - \boldsymbol{\Gamma}_\star\|_{\mathrm{F}}$$

$$\leq C' \frac{\omega(\mathcal{C} \cap \mathbb{S}_F) + u}{\sqrt{n}}\|\boldsymbol{\Gamma}_k - \boldsymbol{\Gamma}_\star\|_{\mathrm{F}}, \tag{42}$$

with probability at least $1 - 2\exp(-u^2)$, where the first inequality is from the fact $\mathcal{R}(\boldsymbol{\Gamma}_k) \leq \mathcal{R}(\boldsymbol{\Gamma}_\star)$, which indicates $(\boldsymbol{\Gamma}_k - \boldsymbol{\Gamma}_\star)/\|\boldsymbol{\Gamma}_k - \boldsymbol{\Gamma}_\star\|_{\mathrm{F}} \in \mathcal{C} \cap \mathbb{S}_F$, the second inequality uses Lemma 4 and the third inequality holds by noting $\mu = 1/\kappa_{\max}$.

The third part III of (38) is the summation of a martingale difference sequence. The difficulty to bound the summation is the coupling between $\{e_t\}$ and $\{x_t\}$. Our analysis to decouple these two sequences is based on the following lemma, which is inspired by [41].

**Lemma 5.** *Under the condition of Theorem 1, if* $n \geq (\omega(\mathcal{C} \cap \mathbb{S}_F) + u)^2$, *we could derive*[2]

$$P\Big(\sup_{\boldsymbol{V} \in \mathcal{C} \cap \mathbb{S}_F} |\frac{1}{n}\langle \boldsymbol{V}, \boldsymbol{X}^T\boldsymbol{E}\rangle| > C\sqrt{\kappa_{\max}}\|\boldsymbol{\Sigma}_e\|^{\frac{1}{2}} \frac{\omega(\mathcal{C} \cap \mathbb{S}_F) + u}{\sqrt{n}}\Big) \leq 2\exp(-u^2). \tag{43}$$

Then we could bound the third part III of (38) with Lemma 5 under the condition of $n \geq (\omega(\mathcal{C} \cap \mathbb{S}_F) + u)^2$ as

$$\sup_{\boldsymbol{V} \in \mathcal{C} \cap \mathbb{S}_F} |\mu\frac{1}{n}\langle \boldsymbol{V}, \boldsymbol{X}^T\boldsymbol{E}\rangle| \leq \mu C\sqrt{\kappa_{\max}}\|\boldsymbol{\Sigma}_e\|^{\frac{1}{2}} \frac{\omega(\mathcal{C} \cap \mathbb{S}_F) + u}{\sqrt{n}}$$

$$\leq C\frac{1}{\sqrt{\kappa_{\max}}}\|\boldsymbol{\Sigma}_e\|^{\frac{1}{2}} \frac{\omega(\mathcal{C} \cap \mathbb{S}_F) + u}{\sqrt{n}}, \tag{44}$$

with probability at least $1 - 2\exp(-u^2)$, where we use $\mu = 1/\kappa_{\max}$.

---

[1]Note this lemma does not only adapt to the random process indexed by two matrices. In fact, from the proof it is explicit that we could derive the deviation bound of the process indexed by any number of matrices if we could handle the increment reformulated by the index matrices. This is why our analysis framework could also be utilized to estimate the regression matrix and the precision matrix simultaneously in [28].

[2]Note this lemma also adapts to the process indexed by more matrices.

Substituting (40), (42) and (44) into (38) yields

$$
\begin{aligned}
&\|\mathbf{\Gamma}_{k+1} - \mathbf{\Gamma}_\star\|_F \\
&\leq (1 - \frac{\kappa_{\min}}{\kappa_{\max}})\|\mathbf{\Gamma}_k - \mathbf{\Gamma}_\star\|_F + C\frac{\omega(\mathcal{C} \cap \mathbb{S}_F) + u}{\sqrt{n}}\|\mathbf{\Gamma}_k - \mathbf{\Gamma}_\star\|_F + C'\frac{1}{\sqrt{\kappa_{\max}}}\|\mathbf{\Sigma}_e\|^{\frac{1}{2}}\frac{\omega(\mathcal{C} \cap \mathbb{S}_F) + u}{\sqrt{n}} \\
&= \underbrace{(1 - \frac{\kappa_{\min}}{\kappa_{\max}} + C\frac{\omega(\mathcal{C} \cap \mathbb{S}_F) + u}{\sqrt{n}})}_{\rho}\|\mathbf{\Gamma}_k - \mathbf{\Gamma}_\star\|_F + \underbrace{C'\frac{1}{\sqrt{\kappa_{\max}}}\|\mathbf{\Sigma}_e\|^{\frac{1}{2}}\frac{\omega(\mathcal{C} \cap \mathbb{S}_F) + u}{\sqrt{n}}}_{\xi} \\
&= \rho\|\mathbf{\Gamma}_k - \mathbf{\Gamma}_\star\|_F + \xi,
\end{aligned}
\tag{45}
$$

with probability at least $1 - 4\exp(-u^2)$, under the condition of $n \geq (2\omega(\mathcal{C} \cap \mathbb{S}_F) + u)^2$.

To guarantee the convergence, that is, $\rho < 1$, we require the number of measurements satisfying

$$
\sqrt{n} > C\frac{\kappa_{\max}}{\kappa_{\min}}(\omega(\mathcal{C} \cap \mathbb{S}_F) + u).
\tag{46}
$$

In this way, we could get the event

$$
\|\mathbf{\Gamma}_{k+1} - \mathbf{\Gamma}_\star\|_F \leq \rho\|\mathbf{\Gamma}_k - \mathbf{\Gamma}_\star\|_F + \xi \leq \rho^{k+1}\|\mathbf{\Gamma}_0 - \mathbf{\Gamma}_\star\|_F + \frac{\xi}{1 - \rho}
\tag{47}
$$

holds with probability at least $1 - c\exp(-u^2)$, where the second inequality is due to taking union bound and the fact $\rho < 1$.

Further, if the number of measurements satisfies

$$
\sqrt{n} > 2C\frac{\kappa_{\max}}{\kappa_{\min}}(\omega(\mathcal{C} \cap \mathbb{S}_F) + u),
\tag{48}
$$

we could derive a simpler expression

$$
\rho = 1 - \frac{\kappa_{\min}}{\kappa_{\max}} + C\frac{\omega(\mathcal{C} \cap \mathbb{S}_F) + u}{\sqrt{n}} < 1 - \frac{\kappa_{\min}}{2\kappa_{\max}}.
\tag{49}
$$

### A.2  Proof of Theorem 2

First we provide several expression used in our analysis.

$$
f_n(\mathbf{S}, \mathbf{L}) = \frac{1}{2n}\|\mathbf{Y} - \mathbf{X}(\mathbf{S} + \mathbf{L})\|_F^2,
\tag{50}
$$

$$
f(\mathbf{S}, \mathbf{L}) = \frac{1}{2}\operatorname{tr}\left\{(\mathbf{S} + \mathbf{L} - \mathbf{S}_\star - \mathbf{L}_\star)^T\mathbf{\Sigma}_x(\mathbf{S} + \mathbf{L} - \mathbf{S}_\star - \mathbf{L}_\star) + \mathbf{I}\right\},
\tag{51}
$$

$$
\nabla_S f_n(\mathbf{S}, \mathbf{L}) = \frac{1}{n}\mathbf{X}^T\mathbf{X}(\mathbf{S} - \mathbf{S}_\star) + \frac{1}{n}\mathbf{X}^T\mathbf{X}(\mathbf{L} - \mathbf{L}_\star) - \frac{1}{n}\mathbf{X}^T\mathbf{E},
\tag{52}
$$

$$
\nabla_L f_n(\mathbf{S}, \mathbf{L}) = \frac{1}{n}\mathbf{X}^T\mathbf{X}(\mathbf{S} - \mathbf{S}_\star) + \frac{1}{n}\mathbf{X}^T\mathbf{X}(\mathbf{L} - \mathbf{L}_\star) - \frac{1}{n}\mathbf{X}^T\mathbf{E},
\tag{53}
$$

$$
\nabla_S f(\mathbf{S}, \mathbf{L}) = \mathbf{\Sigma}_x(\mathbf{S} - \mathbf{S}_\star) + \mathbf{\Sigma}_x(\mathbf{L} - \mathbf{L}_\star),
\tag{54}
$$

$$
\nabla_L f(\mathbf{S}, \mathbf{L}) = \mathbf{\Sigma}_x(\mathbf{S} - \mathbf{S}_\star) + \mathbf{\Sigma}_x(\mathbf{L} - \mathbf{L}_\star).
\tag{55}
$$

In this way, we could bound the difference between $S_{k+1}$ and $S_\star$ as

$$
\begin{aligned}
&\|S_{k+1} - S_\star\|_{\mathrm{F}} \\
&= \|\mathcal{P}_{\mathcal{K}_S}(S_k - \mu\nabla_S f_n(S_k, L_k)) - S_\star\|_{\mathrm{F}} \\
&\leq \sup_{V \in \mathcal{C}_S \cap \mathbb{S}_F} \langle V, S_k - S_\star - \mu\nabla_S f_n(S_k, L_k)\rangle \\
&= \sup_{V \in \mathcal{C}_S \cap \mathbb{S}_F} \langle V, S_k - S_\star - \mu\frac{1}{n}X^T X(S_k - S_\star) - \mu\frac{1}{n}X^T X(L_k - L_\star) + \mu\frac{1}{n}X^T E\rangle \\
&= \sup_{V \in \mathcal{C}_S \cap \mathbb{S}_F} \langle V, (I - \mu\Sigma_x)(S_k - S_\star) - \mu\Sigma_x(L_k - L_\star) \\
&\qquad + \mu(\Sigma_x - \frac{1}{n}X^T X)(S_k - S_\star) + \mu(\Sigma_x - \frac{1}{n}X^T X)(L_k - L_\star) + \mu\frac{1}{n}X^T E\rangle \\
&\leq \underbrace{\sup_{V \in \mathcal{C}_S \cap \mathbb{S}_F} \langle V, (I - \mu\Sigma_x)(S_k - S_\star) - \mu\Sigma_x(L_k - L_\star)\rangle}_{I} \\
&\quad + \underbrace{\sup_{V \in \mathcal{C}_S \cap \mathbb{S}_F} \langle V, \mu(\Sigma_x - \frac{1}{n}X^T X)(S_k - S_\star) + \mu(\Sigma_x - \frac{1}{n}X^T X)(L_k - L_\star)\rangle}_{II} \\
&\quad + \underbrace{\sup_{V \in \mathcal{C}_S \cap \mathbb{S}_F} \langle V, \mu\frac{1}{n}X^T E\rangle}_{III},
\end{aligned}
\tag{56}
$$

where the first inequality refers to Lemma 2 and $\mathcal{C}_S = \mathrm{cone}(\mathcal{K}_S - \{S_\star\})$ is the descent cone generated from $\mathcal{K}_S = \{S \mid \mathcal{R}_S(S) \leq \mathcal{R}_S(S_\star)\}$.

The first term I of (56) only relates to the population loss function $f(S_k, L_k)$ and could be further rearranged as

$$
\begin{aligned}
&\sup_{V \in \mathcal{C}_S \cap \mathbb{S}_F} \langle V, (I - \mu\Sigma_x)(S_k - S_\star) - \mu\Sigma_x(L_k - L_\star)\rangle \\
&\leq \sup_{V \in \mathcal{C}_S \cap \mathbb{S}_F} \langle V, (I - \mu\Sigma_x)(S_k - S_\star)\rangle + \sup_{V \in \mathcal{C}_S \cap \mathbb{S}_F} \langle V, -\mu\Sigma_x(L_k - L_\star)\rangle.
\end{aligned}
\tag{57}
$$

The first term of (57) could be bounded as

$$
\begin{aligned}
\sup_{V \in \mathcal{C}_S \cap \mathbb{S}_F} \langle V, (I - \mu\Sigma_x)(S_k - S_\star)\rangle &\leq \|(I - \mu\Sigma_x)(S_k - S_\star)\|_{\mathrm{F}} \\
&\leq \lambda_{\max}(I - \mu\Sigma_x)\|S_k - S_\star\|_{\mathrm{F}} \\
&\leq (1 - \frac{\kappa_{\min}}{\kappa_{\max}})\|S_k - S_\star\|_{\mathrm{F}},
\end{aligned}
\tag{58}
$$

where the first inequality is due to the Cauchy-Schwartz inequality, the second inequality is from the fact $\|AB\|_{\mathrm{F}} \leq \|A\|\|B\|_{\mathrm{F}}$ for two matrices $A$ and $B$, and the third inequality holds by noting the matrix $I - \mu\Sigma_x$ is positive semi-definite with the largest eigenvalue no more than $1 - \frac{\kappa_{\min}}{\kappa_{\max}}$ under Assumption 2 and the choice of the step size $\mu = 1/\kappa_{\max}$.

The second term of (57) could be bounded as

$$\sup_{\boldsymbol{V}\in\mathcal{C}_S\cap\mathbb{S}_F} \langle \boldsymbol{V}, -\mu\boldsymbol{\Sigma}_x(\boldsymbol{L}_k - \boldsymbol{L}_\star)\rangle$$

$$\leq \sup_{\boldsymbol{V}\in\mathcal{C}_S\cap\mathbb{S}_F, \boldsymbol{U}\in\mathcal{C}_L\cap\mathbb{S}_F} \langle \boldsymbol{V}, -\mu\boldsymbol{\Sigma}_x\boldsymbol{U}\rangle \|\boldsymbol{L}_k - \boldsymbol{L}_\star\|_F$$

$$= \sup_{\boldsymbol{V}\in\mathcal{C}_S\cap\mathbb{S}_F, \boldsymbol{U}\in\mathcal{C}_L\cap\mathbb{S}_F} \langle \mathcal{P}_{\overline{\mathcal{M}}_S}(\boldsymbol{V}) + \mathcal{P}_{\overline{\mathcal{M}}_S^\perp}(\boldsymbol{V}), -\mu\boldsymbol{\Sigma}_x\left(\mathcal{P}_{\overline{\mathcal{M}}_L}(\boldsymbol{U}) + \mathcal{P}_{\overline{\mathcal{M}}_L^\perp}(\boldsymbol{U})\right)\rangle \|\boldsymbol{L}_k - \boldsymbol{L}_\star\|_F$$

$$\leq \mu\Big(\bar{\sigma}_{\max}(\mathcal{P}_{\overline{\mathcal{M}}_S}\boldsymbol{\Sigma}_x\mathcal{P}_{\overline{\mathcal{M}}_L})\|\mathcal{P}_{\overline{\mathcal{M}}_S}(\boldsymbol{V})\|_F\|\mathcal{P}_{\overline{\mathcal{M}}_L}(\boldsymbol{U})\|_F$$

$$+ \bar{\sigma}_{\max}(\mathcal{P}_{\overline{\mathcal{M}}_S}\boldsymbol{\Sigma}_x\mathcal{P}_{\overline{\mathcal{M}}_L^\perp})\|\mathcal{P}_{\overline{\mathcal{M}}_S}(\boldsymbol{V})\|_F\|\mathcal{P}_{\overline{\mathcal{M}}_L^\perp}(\boldsymbol{U})\|_F$$

$$+ \bar{\sigma}_{\max}(\mathcal{P}_{\overline{\mathcal{M}}_S^\perp}\boldsymbol{\Sigma}_x\mathcal{P}_{\overline{\mathcal{M}}_L})\|\mathcal{P}_{\overline{\mathcal{M}}_S^\perp}(\boldsymbol{V})\|_F\|\mathcal{P}_{\overline{\mathcal{M}}_L}(\boldsymbol{U})\|_F$$

$$+ \bar{\sigma}_{\max}(\mathcal{P}_{\overline{\mathcal{M}}_S^\perp}\boldsymbol{\Sigma}_x\mathcal{P}_{\overline{\mathcal{M}}_L^\perp})\|\mathcal{P}_{\overline{\mathcal{M}}_S^\perp}(\boldsymbol{V})\|_F\|\mathcal{P}_{\overline{\mathcal{M}}_L^\perp}(\boldsymbol{U})\|_F\Big)\|\boldsymbol{L}_k - \boldsymbol{L}_\star\|_F$$

$$\leq \mu\frac{\kappa_{\min}}{8}\Big(\|\mathcal{P}_{\overline{\mathcal{M}}_S}(\boldsymbol{V})\|_F\|\mathcal{P}_{\overline{\mathcal{M}}_L}(\boldsymbol{U})\|_F + \|\mathcal{P}_{\overline{\mathcal{M}}_S}(\boldsymbol{V})\|_F\|\mathcal{P}_{\overline{\mathcal{M}}_L^\perp}(\boldsymbol{U})\|_F$$

$$+ \|\mathcal{P}_{\overline{\mathcal{M}}_S^\perp}(\boldsymbol{V})\|_F\|\mathcal{P}_{\overline{\mathcal{M}}_L}(\boldsymbol{U})\|_F + \|\mathcal{P}_{\overline{\mathcal{M}}_S^\perp}(\boldsymbol{V})\|_F\|\mathcal{P}_{\overline{\mathcal{M}}_L^\perp}(\boldsymbol{U})\|_F\Big)\|\boldsymbol{L}_k - \boldsymbol{L}_\star\|_F$$

$$= \mu\frac{\kappa_{\min}}{8}\Big(\|\mathcal{P}_{\overline{\mathcal{M}}_S}(\boldsymbol{V})\|_F + \|\mathcal{P}_{\overline{\mathcal{M}}_S^\perp}(\boldsymbol{V})\|_F\Big)\Big(\|\mathcal{P}_{\overline{\mathcal{M}}_L}(\boldsymbol{U})\|_F + \|\mathcal{P}_{\overline{\mathcal{M}}_L^\perp}(\boldsymbol{U})\|_F\Big)\|\boldsymbol{L}_k - \boldsymbol{L}_\star\|_F$$

$$\leq \mu\frac{\kappa_{\min}}{4}\sqrt{\|\mathcal{P}_{\overline{\mathcal{M}}_S}(\boldsymbol{V})\|_F^2 + \|\mathcal{P}_{\overline{\mathcal{M}}_S^\perp}(\boldsymbol{V})\|_F^2}\sqrt{\|\mathcal{P}_{\overline{\mathcal{M}}_L}(\boldsymbol{U})\|_F^2 + \|\mathcal{P}_{\overline{\mathcal{M}}_L^\perp}(\boldsymbol{U})\|_F^2}\|\boldsymbol{L}_k - \boldsymbol{L}_\star\|_F$$

$$\leq \frac{1}{4}\frac{\kappa_{\min}}{\kappa_{\max}}\|\boldsymbol{L}_k - \boldsymbol{L}_\star\|_F, \tag{59}$$

where the first inequality is from the facts $\mathcal{R}_L(\boldsymbol{L}_k) \leq \mathcal{R}_L(\boldsymbol{L}_\star)$ and $(\boldsymbol{L}_k - \boldsymbol{L}_\star)/\|\boldsymbol{L}_k - \boldsymbol{L}_\star\|_F \in \mathcal{C}_L \cap \mathbb{S}_F$, the second inequality holds by noting the definition of $\bar{\sigma}_{\max}(\cdot)$ and the properties $\mathcal{P}^2 = \mathcal{P}$, $\mathcal{P}^T = \mathcal{P}$ for an orthogonal projection operator $\mathcal{P}$, the third inequality is due to Assumption 3, the second equality is based on $ac + ad + bc + bd = (a+b)(c+d)$ for constant $a, b, c, d$, and the last two inequalities follow from the facts $a + b \leq \sqrt{2}\sqrt{a^2 + b^2}$ for constants $a, b$, $\|\mathcal{P}_{\overline{\mathcal{M}}_S}(\boldsymbol{V})\|_F^2 + \|\mathcal{P}_{\overline{\mathcal{M}}_S^\perp}(\boldsymbol{V})\|_F^2 = \|\boldsymbol{V}\|_F^2 = 1$ and $\|\mathcal{P}_{\overline{\mathcal{M}}_L}(\boldsymbol{U})\|_F^2 + \|\mathcal{P}_{\overline{\mathcal{M}}_L^\perp}(\boldsymbol{U})\|_F^2 = \|\boldsymbol{U}\|_F^2 = 1$.

Substituting (58) and (59) into (57), we could bound the first term I of (56) as

$$\sup_{\boldsymbol{V}\in\mathcal{C}_S\cap\mathbb{S}_F} \langle \boldsymbol{V}, (\boldsymbol{I} - \mu\boldsymbol{\Sigma}_x)(\boldsymbol{S}_k - \boldsymbol{S}_\star) - \mu\boldsymbol{\Sigma}_x(\boldsymbol{L}_k - \boldsymbol{L}_\star)\rangle$$

$$\leq (1 - \frac{\kappa_{\min}}{\kappa_{\max}})\|\boldsymbol{S}_k - \boldsymbol{S}_\star\|_F + \frac{1}{4}\frac{\kappa_{\min}}{\kappa_{\max}}\|\boldsymbol{L}_k - \boldsymbol{L}_\star\|_F. \tag{60}$$

The second part II of (56) contains two terms about the deviation inequality of the sample Gram matrix $\boldsymbol{X}^T\boldsymbol{X}/n$

$$\sup_{\boldsymbol{V}\in\mathcal{C}_S\cap\mathbb{S}_F} \langle \boldsymbol{V}, \mu(\boldsymbol{\Sigma}_x - \frac{1}{n}\boldsymbol{X}^T\boldsymbol{X})(\boldsymbol{S}_k - \boldsymbol{S}_\star) + \mu(\boldsymbol{\Sigma}_x - \frac{1}{n}\boldsymbol{X}^T\boldsymbol{X})(\boldsymbol{L}_k - \boldsymbol{L}_\star)\rangle$$

$$\leq \sup_{\boldsymbol{V}\in\mathcal{C}_S\cap\mathbb{S}_F} \langle \boldsymbol{V}, \mu(\boldsymbol{\Sigma}_x - \frac{1}{n}\boldsymbol{X}^T\boldsymbol{X})(\boldsymbol{S}_k - \boldsymbol{S}_\star)\rangle + \sup_{\boldsymbol{V}\in\mathcal{C}_S\cap\mathbb{S}_F} \langle \boldsymbol{V}, \mu(\boldsymbol{\Sigma}_x - \frac{1}{n}\boldsymbol{X}^T\boldsymbol{X})(\boldsymbol{L}_k - \boldsymbol{L}_\star)\rangle. \tag{61}$$

By setting $\mathcal{C}_1 = \mathcal{C}_2 = \mathcal{C}_S$, we could bound the first term of (61) with Lemma 4 under the condition of $n \geq (2\omega(\mathcal{C}_S \cap \mathbb{S}_F) + u)^2$ as

$$\sup_{\boldsymbol{V}\in\mathcal{C}_S\cap\mathbb{S}_F} \langle \boldsymbol{V}, \mu(\boldsymbol{\Sigma}_x - \frac{1}{n}\boldsymbol{X}^T\boldsymbol{X})(\boldsymbol{S}_k - \boldsymbol{S}_\star)\rangle \leq \sup_{\boldsymbol{U},\boldsymbol{V}\in\mathcal{C}_S\cap\mathbb{S}_F} \langle \boldsymbol{V}, \mu(\boldsymbol{\Sigma}_x - \frac{1}{n}\boldsymbol{X}^T\boldsymbol{X})\boldsymbol{U}\rangle\|\boldsymbol{S}_k - \boldsymbol{S}_\star\|_F$$

$$\leq \mu C\kappa_{\max}\frac{2\omega(\mathcal{C}_S \cap \mathbb{S}_F) + u}{\sqrt{n}}\|\boldsymbol{S}_k - \boldsymbol{S}_\star\|_F$$

$$\leq C\frac{2\omega(\mathcal{C}_S \cap \mathbb{S}_F) + u}{\sqrt{n}}\|\boldsymbol{S}_k - \boldsymbol{S}_\star\|_F, \tag{62}$$

with probability at least $1-2\exp(-u^2)$, where the first inequality is from the fact $\mathcal{R}_S(\boldsymbol{S}_k) \leq \mathcal{R}_S(\boldsymbol{S}_\star)$, which indicates $(\boldsymbol{S}_k - \boldsymbol{S}_\star)/\|\boldsymbol{S}_k - \boldsymbol{S}_\star\|_{\mathrm{F}} \in \mathcal{C}_S \cap \mathbb{S}_F$, the second inequality uses Lemma 4 and the third inequality holds by noting $\mu = 1/\kappa_{\max}$.

By setting $\mathcal{C}_1 = \mathcal{C}_L$ and $\mathcal{C}_2 = \mathcal{C}_S$, we could bound the second term of (61) with Lemma 4 under the condition of $n \geq (\omega(\mathcal{C}_S \cap \mathbb{S}_F) + \omega(\mathcal{C}_L \cap \mathbb{S}_F) + u)^2$ as

$$\sup_{\boldsymbol{V} \in \mathcal{C}_S \cap \mathbb{S}_F} \langle \boldsymbol{V}, \mu(\boldsymbol{\Sigma}_x - \frac{1}{n}\boldsymbol{X}^T\boldsymbol{X})(\boldsymbol{L}_k - \boldsymbol{L}_\star)\rangle$$

$$\leq \sup_{\substack{\boldsymbol{V} \in \mathcal{C}_S \cap \mathbb{S}_F \\ \boldsymbol{U} \in \mathcal{C}_L \cap \mathbb{S}_F}} \langle \boldsymbol{V}, \mu(\boldsymbol{\Sigma}_x - \frac{1}{n}\boldsymbol{X}^T\boldsymbol{X})\boldsymbol{U}\rangle \|\boldsymbol{L}_k - \boldsymbol{L}_\star\|_{\mathrm{F}}$$

$$\leq \mu C \kappa_{\max} \frac{\omega(\mathcal{C}_S \cap \mathbb{S}_F) + \omega(\mathcal{C}_L \cap \mathbb{S}_F) + u}{\sqrt{n}} \|\boldsymbol{L}_k - \boldsymbol{L}_\star\|_{\mathrm{F}}$$

$$\leq C \frac{\omega(\mathcal{C}_S \cap \mathbb{S}_F) + \omega(\mathcal{C}_L \cap \mathbb{S}_F) + u}{\sqrt{n}} \|\boldsymbol{L}_k - \boldsymbol{L}_\star\|_{\mathrm{F}}, \tag{63}$$

with probability at least $1 - 2\exp(-u^2)$, where the first inequality is from the fact $\mathcal{R}_L(\boldsymbol{L}_k) \leq \mathcal{R}_L(\boldsymbol{L}_\star)$, which indicates $(\boldsymbol{L}_k - \boldsymbol{L}_\star)/\|\boldsymbol{L}_k - \boldsymbol{L}_\star\|_{\mathrm{F}} \in \mathcal{C}_L \cap \mathbb{S}_F$, the second inequality uses Lemma 4 and the third inequality holds by noting $\mu = 1/\kappa_{\max}$.

Substituting (62) and (63) into (61), we could bound the second term II of (56) as

$$\sup_{\boldsymbol{V} \in \mathcal{C}_S \cap \mathbb{S}_F} \langle \boldsymbol{V}, \mu(\boldsymbol{\Sigma}_x - \frac{1}{n}\boldsymbol{X}^T\boldsymbol{X})(\boldsymbol{S}_k - \boldsymbol{S}_\star) + \mu(\boldsymbol{\Sigma}_x - \frac{1}{n}\boldsymbol{X}^T\boldsymbol{X})(\boldsymbol{L}_k - \boldsymbol{L}_\star)\rangle$$

$$\leq C \frac{2\omega(\mathcal{C}_S \cap \mathbb{S}_F) + u}{\sqrt{n}} \|\boldsymbol{S}_k - \boldsymbol{S}_\star\|_{\mathrm{F}} + C \frac{\omega(\mathcal{C}_S \cap \mathbb{S}_F) + \omega(\mathcal{C}_L \cap \mathbb{S}_F) + u}{\sqrt{n}} \|\boldsymbol{L}_k - \boldsymbol{L}_\star\|_{\mathrm{F}}, \tag{64}$$

with probability at least $1 - 4\exp(-u^2)$, under the condition of $\sqrt{n} \geq C'''(\omega(\mathcal{C}_S \cap \mathbb{S}_F) + \omega(\mathcal{C}_L \cap \mathbb{S}_F) + u)$.

The third part III of (56) is the summation of a martingale difference sequence, which could be bounded by Lemma 5 under the condition of $n \geq (\omega(\mathcal{C}_S \cap \mathbb{S}_F) + u)^2$

$$\sup_{\boldsymbol{V} \in \mathcal{C}_S \cap \mathbb{S}_F} |\mu \frac{1}{n}\langle \boldsymbol{V}, \boldsymbol{X}^T\boldsymbol{E}\rangle| \leq \mu C' \sqrt{\kappa_{\max}} \|\boldsymbol{\Sigma}_e\|^{\frac{1}{2}} \frac{\omega(\mathcal{C}_S \cap \mathbb{S}_F) + u}{\sqrt{n}}$$

$$\leq C' \frac{1}{\sqrt{\kappa_{\max}}} \|\boldsymbol{\Sigma}_e\|^{\frac{1}{2}} \frac{\omega(\mathcal{C}_S \cap \mathbb{S}_F) + u}{\sqrt{n}}, \tag{65}$$

with probability at least $1 - 2\exp(-u^2)$, where we use $\mu = 1/\kappa_{\max}$.

Substituting (60), (64) and (65) into (56) yields

$$\|\boldsymbol{S}_{k+1} - \boldsymbol{S}_\star\|_{\mathrm{F}}$$

$$\leq (1 - \frac{\kappa_{\min}}{\kappa_{\max}})\|\boldsymbol{S}_k - \boldsymbol{S}_\star\|_{\mathrm{F}} + \frac{1}{4}\frac{\kappa_{\min}}{\kappa_{\max}}\|\boldsymbol{L}_k - \boldsymbol{L}_\star\|_{\mathrm{F}} + C\frac{2\omega(\mathcal{C}_S \cap \mathbb{S}_F) + u}{\sqrt{n}}\|\boldsymbol{S}_k - \boldsymbol{S}_\star\|_{\mathrm{F}}$$

$$+ C\frac{\omega(\mathcal{C}_S \cap \mathbb{S}_F) + \omega(\mathcal{C}_L \cap \mathbb{S}_F) + u}{\sqrt{n}}\|\boldsymbol{L}_k - \boldsymbol{L}_\star\|_{\mathrm{F}} + C'\frac{1}{\sqrt{\kappa_{\max}}}\|\boldsymbol{\Sigma}_e\|^{\frac{1}{2}}\frac{\omega(\mathcal{C}_S \cap \mathbb{S}_F) + u}{\sqrt{n}}, \tag{66}$$

with probability at least $1 - 6\exp(-u^2)$, under the condition of $\sqrt{n} \geq C'''(\omega(\mathcal{C}_S \cap \mathbb{S}_F) + \omega(\mathcal{C}_L \cap \mathbb{S}_F) + u)$.

Following the same procedure, we could derive the bound of $\|\boldsymbol{L}_{k+1} - \boldsymbol{L}_\star\|_{\mathrm{F}}$

$$\|\boldsymbol{L}_{k+1} - \boldsymbol{L}_\star\|_{\mathrm{F}}$$

$$\leq (1 - \frac{\kappa_{\min}}{\kappa_{\max}})\|\boldsymbol{L}_k - \boldsymbol{L}_\star\|_{\mathrm{F}} + \frac{1}{4}\frac{\kappa_{\min}}{\kappa_{\max}}\|\boldsymbol{S}_k - \boldsymbol{S}_\star\|_{\mathrm{F}} + C\frac{2\omega(\mathcal{C}_L \cap \mathbb{S}_F) + u}{\sqrt{n}}\|\boldsymbol{L}_k - \boldsymbol{L}_\star\|_{\mathrm{F}}$$

$$+ C\frac{\omega(\mathcal{C}_L \cap \mathbb{S}_F) + \omega(\mathcal{C}_S \cap \mathbb{S}_F) + u}{\sqrt{n}}\|\boldsymbol{S}_k - \boldsymbol{S}_\star\|_{\mathrm{F}} + C'\frac{1}{\sqrt{\kappa_{\max}}}\|\boldsymbol{\Sigma}_e\|^{\frac{1}{2}}\frac{\omega(\mathcal{C}_L \cap \mathbb{S}_F) + u}{\sqrt{n}}, \tag{67}$$

with probability at least $1 - 6\exp(-u^2)$, under the condition of $\sqrt{n} \geq C'''(\omega(\mathcal{C}_S \cap \mathbb{S}_F) + \omega(\mathcal{C}_L \cap \mathbb{S}_F) + u)$.

Considering (66) and (67) as a whole leads to

$$
\begin{aligned}
&\|\boldsymbol{S}_{k+1} - \boldsymbol{S}_\star\|_{\mathrm{F}} + \|\boldsymbol{L}_{k+1} - \boldsymbol{L}_\star\|_{\mathrm{F}} \\
&\leq \underbrace{(1 - \frac{3}{4}\frac{\kappa_{\min}}{\kappa_{\max}} + C''\frac{\omega(\mathcal{C}_L \cap \mathbb{S}_F) + \omega(\mathcal{C}_S \cap \mathbb{S}_F) + u}{\sqrt{n}})}_{\rho}(\|\boldsymbol{S}_k - \boldsymbol{S}_\star\|_{\mathrm{F}} + \|\boldsymbol{L}_k - \boldsymbol{L}_\star\|_{\mathrm{F}}) \\
&\quad + \underbrace{C'\frac{1}{\sqrt{\kappa_{\max}}}\|\boldsymbol{\Sigma}_e\|^{\frac{1}{2}}\frac{\omega(\mathcal{C}_S \cap \mathbb{S}_F) + \omega(\mathcal{C}_L \cap \mathbb{S}_F) + u}{\sqrt{n}}}_{\xi} \\
&= \rho(\|\boldsymbol{S}_k - \boldsymbol{S}_\star\|_{\mathrm{F}} + \|\boldsymbol{L}_k - \boldsymbol{L}_\star\|_{\mathrm{F}}) + \xi, \quad\quad\quad (68)
\end{aligned}
$$

with probability at least $1 - 12\exp(-u^2)$, under the condition of $\sqrt{n} \geq C'''(\omega(\mathcal{C}_S \cap \mathbb{S}_F) + \omega(\mathcal{C}_L \cap \mathbb{S}_F) + u)$.

To guarantee the convergence, that is, $\rho < 1$, we require the number of measurements satisfying

$$
\sqrt{n} > C''\frac{4}{3}\frac{\kappa_{\max}}{\kappa_{\min}}(\omega(\mathcal{C}_S \cap \mathbb{S}_F) + \omega(\mathcal{C}_L \cap \mathbb{S}_F) + u). \quad\quad\quad (69)
$$

In this way, we could get the event

$$
\begin{aligned}
\|\boldsymbol{S}_{k+1} - \boldsymbol{S}_\star\|_{\mathrm{F}} + \|\boldsymbol{L}_{k+1} - \boldsymbol{L}_\star\|_{\mathrm{F}} &\leq \rho(\|\boldsymbol{S}_k - \boldsymbol{S}_\star\|_{\mathrm{F}} + \|\boldsymbol{L}_k - \boldsymbol{L}_\star\|_{\mathrm{F}}) + \xi \\
&\leq \rho^{k+1}(\|\boldsymbol{S}_0 - \boldsymbol{S}_\star\|_{\mathrm{F}} + \|\boldsymbol{L}_0 - \boldsymbol{L}_\star\|_{\mathrm{F}}) + \frac{\xi}{1 - \rho} \quad (70)
\end{aligned}
$$

holds with probability at least $1 - c\exp(-u^2)$, where the second inequality is due to taking union bound and the fact $\rho < 1$.

Further, if the number of measurements satisfies

$$
\sqrt{n} > 4C''\frac{\kappa_{\max}}{\kappa_{\min}}(\omega(\mathcal{C}_S \cap \mathbb{S}_F) + \omega(\mathcal{C}_L \cap \mathbb{S}_F) + u), \quad\quad\quad (71)
$$

we could derive a simpler format

$$
\rho = 1 - \frac{3}{4}\frac{\kappa_{\min}}{\kappa_{\max}} + C''\frac{\omega(\mathcal{C}_L \cap \mathbb{S}_F) + \omega(\mathcal{C}_S \cap \mathbb{S}_F) + u}{\sqrt{n}} < 1 - \frac{\kappa_{\min}}{2\kappa_{\max}}. \quad\quad\quad (72)
$$

# B  Proof of corollaries

## B.1  Proof of Corollary 1

As the proof of Theorem 1, the difference between $\boldsymbol{\Gamma}_{k+1}$ and $\boldsymbol{\Gamma}_\star$ could be bounded by three parts

$$
\begin{aligned}
&\|\boldsymbol{\Gamma}_{k+1} - \boldsymbol{\Gamma}_\star\|_{\mathrm{F}} \\
&= \|\mathcal{P}_\mathcal{K}(\boldsymbol{\Gamma}_k - \mu\nabla f_n(\boldsymbol{\Gamma}_k)) - \boldsymbol{\Gamma}_\star\|_{\mathrm{F}} \\
&\leq \sup_{\boldsymbol{V} \in \mathcal{C} \cap \mathbb{S}_F} \langle \boldsymbol{V}, \boldsymbol{\Gamma}_k - \boldsymbol{\Gamma}_\star - \mu\nabla f_n(\boldsymbol{\Gamma}_k)\rangle \\
&\leq \underbrace{\sup_{\boldsymbol{V} \in \mathcal{C} \cap \mathbb{S}_F} \langle \boldsymbol{V}, (\boldsymbol{I} - \mu\boldsymbol{\Sigma}_x)(\boldsymbol{\Gamma}_k - \boldsymbol{\Gamma}_\star)\rangle}_{I} + \underbrace{\sup_{\boldsymbol{V} \in \mathcal{C} \cap \mathbb{S}_F} \langle \boldsymbol{V}, \mu(\boldsymbol{\Sigma}_x - \frac{1}{n}\boldsymbol{X}^T\boldsymbol{X})(\boldsymbol{\Gamma}_k - \boldsymbol{\Gamma}_\star)\rangle}_{II} \\
&\quad + \underbrace{\sup_{\boldsymbol{V} \in \mathcal{C} \cap \mathbb{S}_F} \langle \boldsymbol{V}, \mu\frac{1}{n}\boldsymbol{X}^T\boldsymbol{E}\rangle}_{III}. \quad\quad\quad (73)
\end{aligned}
$$

The first term I of (73) about the population loss function could be bounded by the same procedure as the proof of Theorem 1.

When the rows of $\boldsymbol{X}$ are independent with the distribution $\mathcal{N}(\boldsymbol{0}, \boldsymbol{\Sigma}_x)$, the spectral density function becomes $f_x(\theta) = \boldsymbol{\Sigma}_x/(2\pi)$ for any $\theta \in [-\pi, \pi]$ and $\mathcal{M}(f_x) = \lambda_{\max}(\boldsymbol{\Sigma}_x)/(2\pi)$, $m(f_x) = \lambda_{\min}(\boldsymbol{\Sigma}_x)/(2\pi)$. Under this setting, Assumption 2 naturally guarantees

$$\kappa_{\min} \leq \lambda_{\min}(\boldsymbol{\Upsilon}_x) \leq \lambda_{\max}(\boldsymbol{\Upsilon}_x) \leq \kappa_{\max}, \tag{74}$$

$$\kappa_{\min} \leq \lambda_{\min}(\boldsymbol{\Sigma}_x) \leq \lambda_{\max}(\boldsymbol{\Sigma}_x) \leq \kappa_{\max}, \tag{75}$$

where $\boldsymbol{\Upsilon}_x = \boldsymbol{I}_n \otimes \boldsymbol{\Sigma}_x$.

In this way, the conditions of Lemma 4 are satisfied and the second term II of (73) could be bounded following the same procedure as Theorem 1.

With the facts that $\boldsymbol{X}$ and $\boldsymbol{E}$ are independent and their rows are independent Gaussian vectors, we could avoid the decoupling step in the proof of Lemma 5 and derive a similar result except for the absolute constant by following the remained part of the proof for Lemma 5.

Taking the three terms of (73) into consideration, we could derive the result with the same form as Theorem 1.

### B.2 Proof of Corollary 2

In this setting, the objective function is

$$f(\boldsymbol{S}, \boldsymbol{L}) = \frac{1}{2}\|\boldsymbol{Y} - \boldsymbol{S} - \boldsymbol{L}\|_{\mathrm{F}}^2. \tag{76}$$

We could bound the difference between $\boldsymbol{S}_{k+1}$ and $\boldsymbol{S}_\star$ as

$$
\begin{aligned}
&\|\boldsymbol{S}_{k+1} - \boldsymbol{S}_\star\|_{\mathrm{F}} \\
&= \|\mathcal{P}_{\mathcal{K}_S}(\boldsymbol{S}_k - \mu\nabla_S f(\boldsymbol{S}_k, \boldsymbol{L}_k)) - \boldsymbol{S}_\star\|_{\mathrm{F}} \\
&\leq \sup_{\boldsymbol{V} \in \mathcal{C}_S \cap \mathbb{S}_F} \langle \boldsymbol{V}, \boldsymbol{S}_k - \boldsymbol{S}_\star - \mu\nabla_S f(\boldsymbol{S}_k, \boldsymbol{L}_k)\rangle \\
&= \sup_{\boldsymbol{V} \in \mathcal{C}_S \cap \mathbb{S}_F} \langle \boldsymbol{V}, \boldsymbol{S}_k - \boldsymbol{S}_\star - (\boldsymbol{S}_k - \boldsymbol{S}_\star) - (\boldsymbol{L}_k - \boldsymbol{L}_\star) + \boldsymbol{E}\rangle \\
&= \sup_{\boldsymbol{V} \in \mathcal{C}_S \cap \mathbb{S}_F} \langle \boldsymbol{V}, -(\boldsymbol{L}_k - \boldsymbol{L}_\star) + \boldsymbol{E}\rangle \\
&\leq \underbrace{\sup_{\boldsymbol{V} \in \mathcal{C}_S \cap \mathbb{S}_F} \langle \boldsymbol{V}, -(\boldsymbol{L}_k - \boldsymbol{L}_\star)\rangle}_{I} + \underbrace{\sup_{\boldsymbol{V} \in \mathcal{C}_S \cap \mathbb{S}_F} \langle \boldsymbol{V}, \boldsymbol{E}\rangle}_{II},
\end{aligned}
\tag{77}
$$

Based on Assumption 3 and the derivation of (59), we could bound the first term I of (77) as

$$
\begin{aligned}
&\sup_{\boldsymbol{V} \in \mathcal{C}_S \cap \mathbb{S}_F} \langle \boldsymbol{V}, -(\boldsymbol{L}_k - \boldsymbol{L}_\star)\rangle \\
&\leq \sup_{\boldsymbol{V} \in \mathcal{C}_S \cap \mathbb{S}_F, \boldsymbol{U} \in \mathcal{C}_L \cap \mathbb{S}_F} \langle \boldsymbol{V}, -\boldsymbol{U}\rangle \|\boldsymbol{L}_k - \boldsymbol{L}_\star\|_{\mathrm{F}} \\
&= \sup_{\boldsymbol{V} \in \mathcal{C}_S \cap \mathbb{S}_F, \boldsymbol{U} \in \mathcal{C}_L \cap \mathbb{S}_F} \langle \mathcal{P}_{\overline{\mathcal{M}}_S}(\boldsymbol{V}) + \mathcal{P}_{\overline{\mathcal{M}}_S^\perp}(\boldsymbol{V}), -(\mathcal{P}_{\overline{\mathcal{M}}_L}(\boldsymbol{U}) + \mathcal{P}_{\overline{\mathcal{M}}_L^\perp}(\boldsymbol{U}))\rangle \|\boldsymbol{L}_k - \boldsymbol{L}_\star\|_{\mathrm{F}} \\
&\leq \Big( \bar{\sigma}_{\max}(\mathcal{P}_{\overline{\mathcal{M}}_S}\mathcal{P}_{\overline{\mathcal{M}}_L})\|\mathcal{P}_{\overline{\mathcal{M}}_S}(\boldsymbol{V})\|_{\mathrm{F}}\|\mathcal{P}_{\overline{\mathcal{M}}_L}(\boldsymbol{U})\|_{\mathrm{F}} + \bar{\sigma}_{\max}(\mathcal{P}_{\overline{\mathcal{M}}_S}\mathcal{P}_{\overline{\mathcal{M}}_L^\perp})\|\mathcal{P}_{\overline{\mathcal{M}}_S}(\boldsymbol{V})\|_{\mathrm{F}}\|\mathcal{P}_{\overline{\mathcal{M}}_L^\perp}(\boldsymbol{U})\|_{\mathrm{F}} \\
&\qquad + \bar{\sigma}_{\max}(\mathcal{P}_{\overline{\mathcal{M}}_S^\perp}\mathcal{P}_{\overline{\mathcal{M}}_L})\|\mathcal{P}_{\overline{\mathcal{M}}_S^\perp}(\boldsymbol{V})\|_{\mathrm{F}}\|\mathcal{P}_{\overline{\mathcal{M}}_L}(\boldsymbol{U})\|_{\mathrm{F}} \\
&\qquad + \bar{\sigma}_{\max}(\mathcal{P}_{\overline{\mathcal{M}}_S^\perp}\mathcal{P}_{\overline{\mathcal{M}}_L^\perp})\|\mathcal{P}_{\overline{\mathcal{M}}_S^\perp}(\boldsymbol{V})\|_{\mathrm{F}}\|\mathcal{P}_{\overline{\mathcal{M}}_L^\perp}(\boldsymbol{U})\|_{\mathrm{F}} \Big) \|\boldsymbol{L}_k - \boldsymbol{L}_\star\|_{\mathrm{F}} \\
&\leq \frac{1}{8}\Big( \|\mathcal{P}_{\overline{\mathcal{M}}_S}(\boldsymbol{V})\|_{\mathrm{F}}\|\mathcal{P}_{\overline{\mathcal{M}}_L}(\boldsymbol{U})\|_{\mathrm{F}} + \|\mathcal{P}_{\overline{\mathcal{M}}_S}(\boldsymbol{V})\|_{\mathrm{F}}\|\mathcal{P}_{\overline{\mathcal{M}}_L^\perp}(\boldsymbol{U})\|_{\mathrm{F}} \\
&\qquad + \|\mathcal{P}_{\overline{\mathcal{M}}_S^\perp}(\boldsymbol{V})\|_{\mathrm{F}}\|\mathcal{P}_{\overline{\mathcal{M}}_L}(\boldsymbol{U})\|_{\mathrm{F}} + \|\mathcal{P}_{\overline{\mathcal{M}}_S^\perp}(\boldsymbol{V})\|_{\mathrm{F}}\|\mathcal{P}_{\overline{\mathcal{M}}_L^\perp}(\boldsymbol{U})\|_{\mathrm{F}} \Big) \|\boldsymbol{L}_k - \boldsymbol{L}_\star\|_{\mathrm{F}} \\
&= \frac{1}{8}\Big( \|\mathcal{P}_{\overline{\mathcal{M}}_S}(\boldsymbol{V})\|_{\mathrm{F}} + \|\mathcal{P}_{\overline{\mathcal{M}}_S^\perp}(\boldsymbol{V})\|_{\mathrm{F}} \Big)\Big( \|\mathcal{P}_{\overline{\mathcal{M}}_L}(\boldsymbol{U})\|_{\mathrm{F}} + \|\mathcal{P}_{\overline{\mathcal{M}}_L^\perp}(\boldsymbol{U})\|_{\mathrm{F}} \Big) \|\boldsymbol{L}_k - \boldsymbol{L}_\star\|_{\mathrm{F}} \\
&\leq \frac{1}{4}\sqrt{\|\mathcal{P}_{\overline{\mathcal{M}}_S}(\boldsymbol{V})\|_{\mathrm{F}}^2 + \|\mathcal{P}_{\overline{\mathcal{M}}_S^\perp}(\boldsymbol{V})\|_{\mathrm{F}}^2}\sqrt{\|\mathcal{P}_{\overline{\mathcal{M}}_L}(\boldsymbol{U})\|_{\mathrm{F}}^2 + \|\mathcal{P}_{\overline{\mathcal{M}}_L^\perp}(\boldsymbol{U})\|_{\mathrm{F}}^2}\|\boldsymbol{L}_k - \boldsymbol{L}_\star\|_{\mathrm{F}} \\
&\leq \frac{1}{4}\|\boldsymbol{L}_k - \boldsymbol{L}_\star\|_{\mathrm{F}}.
\end{aligned}
\tag{78}
$$

Under the robust PCA setting, the noise matrix $\boldsymbol{E}$ could be interpreted as

$$\boldsymbol{E} = \frac{1}{n}\sum_{i=1}^{n} \boldsymbol{z}_i \boldsymbol{z}_i^T - (\boldsymbol{S}_\star + \boldsymbol{L}_\star) = \frac{1}{n}\boldsymbol{Z}^T\boldsymbol{Z} - (\boldsymbol{S}_\star + \boldsymbol{L}_\star), \tag{79}$$

where $\{\boldsymbol{z}_i\}$ is a sequence of independent Gaussian vectors with the distribution $\mathcal{N}(\boldsymbol{0}, \boldsymbol{S}_\star + \boldsymbol{L}_\star)$, $\boldsymbol{Z} = [\boldsymbol{z}_1, \cdots, \boldsymbol{z}_n]^T$ is a matrix whose rows are i.i.d. Gaussian vectors and $\mathbb{E}[\frac{1}{n}\boldsymbol{Z}^T\boldsymbol{Z}] = \boldsymbol{S}_\star + \boldsymbol{L}_\star$.

In this way, the second term II of (77) could be bounded as the second term II of (73).

Taking the two parts of (77) into consideration, we could derive

$$\|\boldsymbol{S}_{k+1} - \boldsymbol{S}_\star\|_{\mathrm{F}} \le \frac{1}{4}\|\boldsymbol{L}_k - \boldsymbol{L}_\star\|_{\mathrm{F}} + C\|\boldsymbol{S}_\star + \boldsymbol{L}_\star\|\frac{\omega(\mathcal{C}_S \cap \mathbb{S}_F) + u}{\sqrt{n}}, \tag{80}$$

with probability at least $1 - 2\exp(-u^2)$, under the condition $\sqrt{n} > \omega(\mathcal{C}_S \cap \mathbb{S}_F) + u$.

The difference between $\boldsymbol{L}_{k+1}$ and $\boldsymbol{L}_\star$ could be derived with the same way.

From [42, Corollary 10.3.4 and Exercise 10.4.4], we could derive $\omega(\mathcal{C}_S \cap \mathbb{S}_F) \le C'\sqrt{s_\star \log d}$ and $\omega(\mathcal{C}_L \cap \mathbb{S}_F) \le C''\sqrt{r_\star d}$.

## C  Proof of auxiliary lemmas

### C.1  Proof of Lemma 2

From the definition of projection, $\bar{\boldsymbol{x}}$ is the optimal solution of the following optimization problem

$$\bar{\boldsymbol{x}} = \operatorname*{argmin}_{\boldsymbol{x}} \iota_{\mathcal{K}}(\boldsymbol{x}) + \frac{1}{2}\|\boldsymbol{x} - \boldsymbol{y}\|_2^2, \tag{81}$$

where $\iota_{\mathcal{K}}(\cdot)$ is the indicator function defined as

$$\iota_{\mathcal{K}}(\boldsymbol{x}) = \begin{cases} 0 & \text{if } \boldsymbol{x} \in \mathcal{K}, \\ \infty & \text{otherwise.} \end{cases} \tag{82}$$

According to the fact that $\bar{\boldsymbol{x}}$ is the optimal solution, we could derive

$$\boldsymbol{0} \in \partial\iota_{\mathcal{K}}(\bar{\boldsymbol{x}}) + \bar{\boldsymbol{x}} - \boldsymbol{y} = \partial\iota_{\mathcal{K}}(\bar{\boldsymbol{x}}) + \bar{\boldsymbol{x}} - \boldsymbol{x}^\star + \boldsymbol{x}^\star - \boldsymbol{y}, \tag{83}$$

where $\partial\iota_{\mathcal{K}}(\bar{\boldsymbol{x}})$ is the subdifferential of $\iota_{\mathcal{K}}(\cdot)$ at $\bar{\boldsymbol{x}}$.

After reformulation, we could derive

$$-(\bar{\boldsymbol{x}} - \boldsymbol{x}^\star + \boldsymbol{x}^\star - \boldsymbol{y}) \in \partial\iota_{\mathcal{K}}(\bar{\boldsymbol{x}}) = N(\bar{\boldsymbol{x}}; \mathcal{K}), \tag{84}$$

where $N(\bar{\boldsymbol{x}}; \mathcal{K})$ is the normal cone of $\mathcal{K}$ at $\bar{\boldsymbol{x}}$. Here we adopt the fact that $\partial\iota_{\mathcal{K}}(\bar{\boldsymbol{x}}) = N(\bar{\boldsymbol{x}}; \mathcal{K})$ from [43, Example 2.32] and the normal cone at $\bar{\boldsymbol{x}} \in \mathcal{K}$ is defined in [43, Definition 9] as

$$N(\bar{\boldsymbol{x}}; \mathcal{K}) := \{\boldsymbol{v} \mid \langle \boldsymbol{v}, \boldsymbol{x} - \bar{\boldsymbol{x}}\rangle \le 0, \ \forall \boldsymbol{x} \in \mathcal{K}\}. \tag{85}$$

Combining with the definition of normal cone (85), we could get

$$\langle -(\bar{\boldsymbol{x}} - \boldsymbol{x}^\star + \boldsymbol{x}^\star - \boldsymbol{y}), \boldsymbol{x}^\star - \bar{\boldsymbol{x}}\rangle \le 0, \tag{86}$$

where we use the fact $\boldsymbol{x}^\star \in \mathcal{K}$.

Then it is easy to verify that

$$\|\bar{\boldsymbol{x}} - \boldsymbol{x}^\star\|_2^2 \le \langle \bar{\boldsymbol{x}} - \boldsymbol{x}^\star, \boldsymbol{y} - \boldsymbol{x}^\star\rangle \le \sup_{\boldsymbol{v} \in \mathcal{C} \cap \mathbb{S}_2} \langle \boldsymbol{v}, \boldsymbol{y} - \boldsymbol{x}^\star\rangle \|\bar{\boldsymbol{x}} - \boldsymbol{x}^\star\|_2, \tag{87}$$

where the second inequality is from $(\bar{\boldsymbol{x}} - \boldsymbol{x}^\star)/\|\bar{\boldsymbol{x}} - \boldsymbol{x}^\star\|_2 \in \mathcal{C} \cap \mathbb{S}_2$.

## C.2 Proof of Lemma 3

First, we write $\mathbf{\Gamma} - \mathbf{\Gamma}_\star - \mu \nabla f(\mathbf{\Gamma})$ as

$$\mathbf{\Gamma} - \mathbf{\Gamma}_\star - \mu \nabla f(\mathbf{\Gamma}) = (\mathbf{I} - \mu \mathbf{\Sigma}_x)(\mathbf{\Gamma} - \mathbf{\Gamma}_\star). \tag{88}$$

Under Assumption 2 and the choice of step size $\mu = 1/\kappa_{\max}$, it is easy to verify that the matrix $\mathbf{I} - \mu \mathbf{\Sigma}_x$ is positive semi-definite and its largest eigenvalue is no more than $1 - \frac{\kappa_{\min}}{\kappa_{\max}}$. Then we could derive

$$\|\mathbf{\Gamma} - \mathbf{\Gamma}_\star - \mu \nabla f(\mathbf{\Gamma})\|_{\mathrm{F}} \leq \lambda_{\max}(\mathbf{I} - \mu \mathbf{\Sigma}_x)\|\mathbf{\Gamma} - \mathbf{\Gamma}_\star\|_{\mathrm{F}} \leq (1 - \frac{\kappa_{\min}}{\kappa_{\max}})\|\mathbf{\Gamma} - \mathbf{\Gamma}_\star\|_{\mathrm{F}}, \tag{89}$$

where we use the fact that $\|\mathbf{AB}\|_{\mathrm{F}} \leq \|\mathbf{A}\|\|\mathbf{B}\|_{\mathrm{F}}$ for two matrices $\mathbf{A}$ and $\mathbf{B}$.

## C.3 Proof of Lemma 4

To bound the supremum of the random process $X_{\mathbf{U},\mathbf{V}} = \langle \mathbf{V}, (\mathbf{\Sigma}_x - \frac{\mathbf{X}^T\mathbf{X}}{n})\mathbf{U}\rangle$, where $\mathbf{U} \in \mathcal{C}_1 \cap \mathbb{S}_F$ and $\mathbf{V} \in \mathcal{C}_2 \cap \mathbb{S}_F$, we first illustrate the random process $X_{\mathbf{U},\mathbf{V}}$ has a mixed tail and then apply Lemma 6.

**Lemma 6.** *[44, Theorem 3.5] Suppose the random process $(X_t)_{t \in T}$ has a mixed tail*

$$P(|X_t - X_s| > u) \leq 2\exp\left(-\min(\frac{u^2}{d_2(t,s)^2}, \frac{u}{d_1(t,s)})\right), \tag{90}$$

*then we could derive*

$$P\left(\sup_{t \in T}|X_t - X_{t_0}| > C\left(\gamma_2(T, d_2) + \gamma_1(T, d_1) + u\Delta_2(T) + u^2\Delta_1(T)\right)\right) \leq 2\exp(-u^2), \tag{91}$$

*where $\Delta_2(T)$ $(\Delta_1(T))$ is the diameter of $T$ with respect to the semi-metric $d_2$ $(d_1)$.*

Here $\gamma_\alpha$-functional is defined as [45, Section 2.3]

$$\gamma_\alpha(T, d) = \inf_{\mathcal{T}} \sup_{t \in T} \sum_{n=0}^{\infty} 2^{n/\alpha} d(t, T_n),$$

for $0 < \alpha < \infty$.

First we could rearrange the increment as

$$
\begin{aligned}
X_{\mathbf{U},\mathbf{V}} &- X_{\mathbf{W},\mathbf{Z}} \\
&= \langle \mathbf{V}, (\mathbf{\Sigma}_x - \frac{\mathbf{X}^T\mathbf{X}}{n})\mathbf{U}\rangle - \langle \mathbf{Z}, (\mathbf{\Sigma}_x - \frac{\mathbf{X}^T\mathbf{X}}{n})\mathbf{W}\rangle \\
&= \operatorname{tr}\left\{(\mathbf{\Sigma}_x - \frac{\mathbf{X}^T\mathbf{X}}{n})(\mathbf{U}\mathbf{V}^T - \mathbf{W}\mathbf{Z}^T)\right\} \\
&= \mathbb{E}\left[\frac{1}{n}\operatorname{vec}(\mathbf{X}^T)^T(\mathbf{I}_n \otimes (\mathbf{U}\mathbf{V}^T - \mathbf{W}\mathbf{Z}^T))\operatorname{vec}(\mathbf{X}^T)\right] \\
&\quad - \frac{1}{n}\operatorname{vec}(\mathbf{X}^T)^T(\mathbf{I}_n \otimes (\mathbf{U}\mathbf{V}^T - \mathbf{W}\mathbf{Z}^T))\operatorname{vec}(\mathbf{X}^T).
\end{aligned}
\tag{92}
$$

We could further rearrange $\mathbf{U}\mathbf{V}^T - \mathbf{W}\mathbf{Z}^T$ as

$$\mathbf{U}\mathbf{V}^T - \mathbf{W}\mathbf{Z}^T = \mathbf{U}\mathbf{V}^T - \mathbf{W}\mathbf{V}^T + \mathbf{W}\mathbf{V}^T - \mathbf{W}\mathbf{Z}^T = (\mathbf{U} - \mathbf{W})\mathbf{V}^T + \mathbf{W}(\mathbf{V} - \mathbf{Z})^T. \tag{93}$$

Its Frobenius norm could be bounded as

$$\|\mathbf{U}\mathbf{V}^T - \mathbf{W}\mathbf{Z}^T\|_{\mathrm{F}}^2 \leq 2\|\mathbf{U} - \mathbf{W}\|_{\mathrm{F}}^2 + 2\|\mathbf{V} - \mathbf{Z}\|_{\mathrm{F}}^2 \leq 2\|\left(\begin{smallmatrix}\mathbf{U}\\\mathbf{V}\end{smallmatrix}\right) - \left(\begin{smallmatrix}\mathbf{W}\\\mathbf{Z}\end{smallmatrix}\right)\|_{\mathrm{F}}^2. \tag{94}$$

The following lemma illustrates the quadratic form $\langle \mathbf{U}^T, \mathbf{X}^T\mathbf{X}\rangle$ has a mixed tail.

**Lemma 7.** *Suppose $vec(\boldsymbol{X}^T)$ follows the distribution $\mathcal{N}(\boldsymbol{0}, \boldsymbol{\Upsilon}_x)$. We have the tail bound*

$$P\Big(|\operatorname{tr}(\boldsymbol{X}\boldsymbol{U}\boldsymbol{X}^T) - \mathbb{E}\operatorname{tr}(\boldsymbol{X}\boldsymbol{U}\boldsymbol{X}^T)| > u\Big) \leq 2\exp\Big(-c\min(\frac{u^2}{n\|\boldsymbol{\Upsilon}_x\|^2\|\boldsymbol{U}\|_{\mathrm{F}}^2}, \frac{u}{\|\boldsymbol{\Upsilon}_x\|\|\boldsymbol{U}\|_{\mathrm{F}}})\Big), \quad (95)$$

*where $c$ is a constant.*

For the VAR model (1), we could verify $vec(\boldsymbol{X}^T) \sim \mathcal{N}(\boldsymbol{0}, \boldsymbol{\Upsilon}_x)$ and $\mathbb{E}\boldsymbol{X}^T\boldsymbol{X} = n\boldsymbol{\Sigma}_x$. In this way, the conditions of Lemma 7 are satisfied. Then we could derive

$$P\Big(|\langle \boldsymbol{V}, (\boldsymbol{\Sigma}_x - \frac{\boldsymbol{X}^T\boldsymbol{X}}{n})\boldsymbol{U}\rangle - \langle \boldsymbol{Z}, (\boldsymbol{\Sigma}_x - \frac{\boldsymbol{X}^T\boldsymbol{X}}{n})\boldsymbol{W}\rangle| > u\Big)$$

$$\leq 2\exp\Big(-c\min(\frac{u^2}{\frac{2}{n}\|\boldsymbol{\Upsilon}_x\|^2\|(\begin{smallmatrix}\boldsymbol{U}\\\boldsymbol{V}\end{smallmatrix}) - (\begin{smallmatrix}\boldsymbol{W}\\\boldsymbol{Z}\end{smallmatrix})\|_{\mathrm{F}}^2}, \frac{u}{\frac{\sqrt{2}}{n}\|\boldsymbol{\Upsilon}_x\|\|(\begin{smallmatrix}\boldsymbol{U}\\\boldsymbol{V}\end{smallmatrix}) - (\begin{smallmatrix}\boldsymbol{W}\\\boldsymbol{Z}\end{smallmatrix})\|_{\mathrm{F}}})\Big)$$

$$\leq 2\exp\Big(-c\min(\frac{u^2}{\frac{2}{n}\kappa_{\max}^2\|(\begin{smallmatrix}\boldsymbol{U}\\\boldsymbol{V}\end{smallmatrix}) - (\begin{smallmatrix}\boldsymbol{W}\\\boldsymbol{Z}\end{smallmatrix})\|_{\mathrm{F}}^2}, \frac{u}{\frac{\sqrt{2}}{n}\kappa_{\max}\|(\begin{smallmatrix}\boldsymbol{U}\\\boldsymbol{V}\end{smallmatrix}) - (\begin{smallmatrix}\boldsymbol{W}\\\boldsymbol{Z}\end{smallmatrix})\|_{\mathrm{F}}})\Big)$$

$$\leq 2\exp\Big(-\min(\frac{u^2}{\frac{2}{n}\kappa_{\max}^2 C^2\|(\begin{smallmatrix}\boldsymbol{U}\\\boldsymbol{V}\end{smallmatrix}) - (\begin{smallmatrix}\boldsymbol{W}\\\boldsymbol{Z}\end{smallmatrix})\|_{\mathrm{F}}^2}, \frac{u}{\frac{\sqrt{2}}{n}\kappa_{\max} C\|(\begin{smallmatrix}\boldsymbol{U}\\\boldsymbol{V}\end{smallmatrix}) - (\begin{smallmatrix}\boldsymbol{W}\\\boldsymbol{Z}\end{smallmatrix})\|_{\mathrm{F}}})\Big), \quad (96)$$

where the second inequality is from the condition $\lambda_{\max}(\boldsymbol{\Upsilon}_x) \leq \kappa_{\max}$ and the last inequality holds for two positive constants $c \leq 1$ and $C \geq 1$.

From (96), the increment $X_{\boldsymbol{U},\boldsymbol{V}} - X_{\boldsymbol{W},\boldsymbol{Z}}$ has a mixed tail with $d_2 = \sqrt{2}C\kappa_{\max}\|\cdot\|_{\mathrm{F}}/\sqrt{n}$ and $d_1 = \sqrt{2}C\kappa_{\max}\|\cdot\|_{\mathrm{F}}/n$.

Combined with Lemma 6, we could derive the event

$$\sup_{\substack{\boldsymbol{U}\in\mathcal{C}_1\cap\mathbb{S}_F\\\boldsymbol{V}\in\mathcal{C}_2\cap\mathbb{S}_F}} |\langle \boldsymbol{V}, (\boldsymbol{\Sigma}_x - \frac{\boldsymbol{X}^T\boldsymbol{X}}{n})\boldsymbol{U}\rangle| > C'\Big(\gamma_2(T, d_2) + \gamma_1(T, d_1) + u\Delta_2(T) + u^2\Delta_1(T)\Big) \quad (97)$$

holds with probability at most $2\exp(-u^2)$. Here $T = \mathcal{C}_1 \cap \mathbb{S}_F \times \mathcal{C}_2 \cap \mathbb{S}_F$.

We adopt the following lemma to transfer the $\gamma_1$-functional to the $\gamma_2$-functional and deal with the coefficients of metrics.

**Lemma 8.** *[21, Lemma 2.7, Equation 46] For $\gamma_\alpha$-functional, we have*

$$\gamma_1(S, \|\cdot\|_2) \leq \gamma_2^2(S, \|\cdot\|_2), \quad (98)$$
$$\gamma_\alpha(S, cd) = c\gamma_\alpha(S, d), \quad (99)$$

*where $\alpha > 0$, $c > 0$.*

Combining with the Talagrand's majorizing measure theorem [46, Theorem 2.1.1], we could bound the $\gamma_2$-functional by the Gaussian width

$$\gamma_2(T, \|\cdot\|_{\mathrm{F}}) \leq C''\omega(T) \leq C''(\omega(\mathcal{C}_1 \cap \mathbb{S}_F) + \omega(\mathcal{C}_2 \cap \mathbb{S}_F)), \quad (100)$$

where the Frobenius norm for a matrix is equivalent to the $l_2$ norm for a vector.

Then we could rearrange (97) further with Lemma 8 and (100)

$$\sup_{\substack{\boldsymbol{U}\in\mathcal{C}_1\cap\mathbb{S}_F\\\boldsymbol{V}\in\mathcal{C}_2\cap\mathbb{S}_F}} |\langle \boldsymbol{V}, (\boldsymbol{\Sigma}_x - \frac{\boldsymbol{X}^T\boldsymbol{X}}{n})\boldsymbol{U}\rangle|$$

$$> C'\Big(\sqrt{2}CC''\kappa_{\max}\frac{\omega(\mathcal{C}_1 \cap \mathbb{S}_F) + \omega(\mathcal{C}_2 \cap \mathbb{S}_F)}{\sqrt{n}} + \sqrt{2}C(C'')^2\kappa_{\max}\frac{(\omega(\mathcal{C}_1 \cap \mathbb{S}_F) + \omega(\mathcal{C}_2 \cap \mathbb{S}_F))^2}{n}$$

$$+ \sqrt{2}C\kappa_{\max}\frac{u}{\sqrt{n}}\Delta_F(T) + \sqrt{2}C\kappa_{\max}\frac{u^2}{n}\Delta_F(T)\Big) \quad (101)$$

holds with probability at most $2\exp(-u^2)$, where we use the facts $\Delta_2(T) = \sqrt{2}C\kappa_{\max}\Delta_F(T)/\sqrt{n}$ and $\Delta_1(T) = \sqrt{2}C\kappa_{\max}\Delta_F(T)/n$.

From the facts $(\omega(\mathcal{C}_1 \cap \mathbb{S}_F) + \omega(\mathcal{C}_2 \cap \mathbb{S}_F))^2 + u^2 \le (\omega(\mathcal{C}_1 \cap \mathbb{S}_F) + \omega(\mathcal{C}_2 \cap \mathbb{S}_F) + u)^2$ and $\Delta_F(T) \le 4$, we could rearrange (97) when the item $(\omega(\mathcal{C}_1 \cap \mathbb{S}_F) + \omega(\mathcal{C}_2 \cap \mathbb{S}_F) + u)/\sqrt{n}$ is dominant and derive

$$P\left(\sup_{\substack{\boldsymbol{U} \in \mathcal{C}_1 \cap \mathbb{S}_F \\ \boldsymbol{V} \in \mathcal{C}_2 \cap \mathbb{S}_F}} |\langle \boldsymbol{V}, (\boldsymbol{\Sigma}_x - \frac{\boldsymbol{X}^T \boldsymbol{X}}{n}) \boldsymbol{U} \rangle| > C''' \kappa_{\max} \frac{\omega(\mathcal{C}_1 \cap \mathbb{S}_F) + \omega(\mathcal{C}_2 \cap \mathbb{S}_F) + u}{\sqrt{n}}\right) \le 2 \exp(-u^2),$$

(102)

when $n \ge (\omega(\mathcal{C}_1 \cap \mathbb{S}_F) + \omega(\mathcal{C}_2 \cap \mathbb{S}_F) + u)^2$.

### C.4 Proof of Lemma 7

This lemma is a direct corollary of the Hanson-Wright inequality.

**Lemma 9** (Hanson-Wright inequality [47])**.** *Suppose $\boldsymbol{x}$ is a random vector with independent sub-Gaussian components $\boldsymbol{x}_i$ satisfying $\mathbb{E}[\boldsymbol{x}_i] = 0$ and $\|\boldsymbol{x}_i\|_{\psi_2} \le K$. $\boldsymbol{A} \in \mathbb{R}^{n \times n}$ is a fixed matrix. For $u > 0$, we could get*

$$P(|\boldsymbol{x}^T \boldsymbol{A} \boldsymbol{x} - \mathbb{E}\boldsymbol{x}^T \boldsymbol{A} \boldsymbol{x}| > u) \le 2 \exp\left(-c \min(-\frac{u^2}{K^4 \|\boldsymbol{A}\|_{\mathrm{F}}^2}, \frac{u}{K^2 \|\boldsymbol{A}\|})\right),$$

(103)

*where $c > 0$ is a constant.*

First, we could rearrange

$$\mathrm{tr}(\boldsymbol{X} \boldsymbol{U} \boldsymbol{X}^T) = \mathrm{vec}(\boldsymbol{X}^T)^T (\boldsymbol{I}_n \otimes \boldsymbol{U}) \mathrm{vec}(\boldsymbol{X}^T) = \mathrm{vec}(\boldsymbol{X}^T)^T \boldsymbol{\Upsilon}_x^{-\frac{1}{2}} \boldsymbol{\Upsilon}_x^{\frac{1}{2}} (\boldsymbol{I}_n \otimes \boldsymbol{U}) \boldsymbol{\Upsilon}_x^{\frac{1}{2}} \boldsymbol{\Upsilon}_x^{-\frac{1}{2}} \mathrm{vec}(\boldsymbol{X}^T).$$

(104)

In this way, $\boldsymbol{\Upsilon}_x^{-\frac{1}{2}} \mathrm{vec}(\boldsymbol{X}^T)$ becomes an isotropic Gaussian vector. Combining the rotation invariance of Gaussian vectors, we could derive

$$P\left(|\mathrm{tr}(\boldsymbol{X} \boldsymbol{U} \boldsymbol{X}^T) - \mathbb{E}\mathrm{tr}(\boldsymbol{X} \boldsymbol{U} \boldsymbol{X}^T)| > u\right)$$

$$= P\left(|\boldsymbol{g}^T \boldsymbol{\Upsilon}_x^{\frac{1}{2}} (\boldsymbol{I}_n \otimes \boldsymbol{U}) \boldsymbol{\Upsilon}_x^{\frac{1}{2}} \boldsymbol{g} - \mathbb{E}\boldsymbol{g}^T \boldsymbol{\Upsilon}_x^{\frac{1}{2}} (\boldsymbol{I}_n \otimes \boldsymbol{U}) \boldsymbol{\Upsilon}_x^{\frac{1}{2}} \boldsymbol{g}| > u\right)$$

$$\le 2 \exp\left(-c \min(\frac{u^2}{\|\boldsymbol{\Upsilon}_x^{\frac{1}{2}} (\boldsymbol{I}_n \otimes \boldsymbol{U}) \boldsymbol{\Upsilon}_x^{\frac{1}{2}}\|_{\mathrm{F}}^2}, \frac{u}{\|\boldsymbol{\Upsilon}_x^{\frac{1}{2}} (\boldsymbol{I}_n \otimes \boldsymbol{U}) \boldsymbol{\Upsilon}_x^{\frac{1}{2}}\|})\right)$$

$$\le 2 \exp\left(-c \min(\frac{u^2}{n\|\boldsymbol{\Upsilon}_x\|^2 \|\boldsymbol{U}\|_{\mathrm{F}}^2}, \frac{u}{\|\boldsymbol{\Upsilon}_x\| \|\boldsymbol{U}\|_{\mathrm{F}}})\right),$$

where $\boldsymbol{g}$ is a vector with independent standard Gaussian entries. Here we use $\|\boldsymbol{A}\boldsymbol{B}\|_{\mathrm{F}} \le \|\boldsymbol{A}\| \|\boldsymbol{B}\|_{\mathrm{F}}$, $\|\boldsymbol{A}\boldsymbol{B}\| \le \|\boldsymbol{A}\| \|\boldsymbol{B}\|$ and $\|\boldsymbol{A}\| \le \|\boldsymbol{A}\|_{\mathrm{F}}$ for two matrices $\boldsymbol{A}$ and $\boldsymbol{B}$.

### C.5 Proof of Lemma 5

The analysis is inspired by [41, Theorem 1], which is based on the decoupling theory in [48]. First we introduce two related definitions.

**Definition 2** ($\{\mathcal{F}_i\}$-tangent sequence)**.** *[48, Definition 2.1] Let $\{d_i\}$ be a sequence of random variables adapted to an increasing sequence of $\sigma$-fields $\{\mathcal{F}_i\}$ and assume $\mathcal{F}_0$ is the trivial $\sigma$-field. Then a sequence $\{e_i\}$ adapted to $\{\mathcal{F}_i\}$ is $\{\mathcal{F}_i\}$-tangent to $\{d_i\}$ if for all $i$*

$$p(d_i|\mathcal{F}_{i-1}) = p(e_i|\mathcal{F}_{i-1}).$$

(105)

**Definition 3** (Decoupled sequence)**.** *[48, Definition 2.2] A sequence $\{e_i\}$ of random variables adapted to an increasing sequence of $\sigma$-fields $\{\mathcal{F}_i\}$ contained in $\mathcal{F}$ is said to satisfy condition CI if there exists a $\sigma$-algebra $\mathcal{G}$ contained in $\mathcal{F}$ such that $\{e_i\}$ is a sequence of conditionally independent random variables given $\mathcal{G}$ and*

$$p(d_i|\mathcal{F}_{i-1}) = p(e_i|\mathcal{F}_{i-1}) = p(e_i|\mathcal{G})$$

*for all $i$. Then the sequence $\{e_i\}$ is said to be decoupled.*

The following lemma guarantees the existence of the decoupled sequence.

**Lemma 10.** *[48] For any sequence $\{d_i\}$ adapted to an increasing sequence of $\sigma$-fields $\{\mathcal{F}_i\}$, there alway exists a sequence $\{e_i\}$, which is $\{\mathcal{F}_i\}$-tangent to $\{d_i\}$ and satisfies the CI condition.*

The following lemma is an important tool to analyze the exponential inequalities for martingales.

**Lemma 11.** *[48, Corollary 3.1] Let $\{d_i\}$, $\{e_i\}$ be $\{\mathcal{F}_i\}$-tangent. Suppose $\{e_i\}$ is decoupled. Let $g \geq 0$ be any random variable measurable respect to $\sigma(\{d_i\}_{i=1}^\infty)$. Then for any finite $t$,*

$$\mathbb{E}[g \exp(t \sum_{i=1}^n d_i)] \leq \sqrt{\mathbb{E}[g^2 \exp(2t \sum_{i=1}^n e_i)]}. \tag{106}$$

The core of the proof is to illustrate the increment $\langle \boldsymbol{V} - \boldsymbol{Z}, \boldsymbol{X}^T \boldsymbol{E} \rangle$ has a mixed tail.

First, we could rearrange $\langle \boldsymbol{V} - \boldsymbol{Z}, \boldsymbol{X}^T \boldsymbol{E} \rangle$ as

$$\langle \boldsymbol{V} - \boldsymbol{Z}, \boldsymbol{X}^T \boldsymbol{E} \rangle = \sum_{t=1}^n \boldsymbol{e}_t^T (\boldsymbol{V} - \boldsymbol{Z})^T \boldsymbol{x}_{t-1} = \sum_{t=1}^n m_t, \tag{107}$$

where the term of the martingale difference sequence $\{m_t\}$ is defined as $m_t = \boldsymbol{e}_t^T (\boldsymbol{V} - \boldsymbol{Z})^T \boldsymbol{x}_{t-1}$. Further $m_t$ could be viewed as a zero-mean Gaussian variable with the variance $\|\boldsymbol{\Sigma}_e^{\frac{1}{2}}(\boldsymbol{V} - \boldsymbol{Z})^T \boldsymbol{x}_{t-1}\|_2^2$ conditioned on $\mathcal{F}_{t-1} = \sigma(\{\boldsymbol{e}_0, \cdots, \boldsymbol{e}_{t-1}\})$, where we set $\boldsymbol{e}_0 = \boldsymbol{x}_0$ for we consider the stationary data $\{\boldsymbol{x}_t\}_{t=0}^n$. We could rewrite the moment generating function (MGF) of $\langle \boldsymbol{V} - \boldsymbol{Z}, \boldsymbol{X}^T \boldsymbol{E} \rangle$ as

$$\mathbb{E}[\exp(\lambda \sum_{t=1}^n \boldsymbol{e}_t^T (\boldsymbol{V} - \boldsymbol{Z})^T \boldsymbol{x}_{t-1})]$$

$$= \mathbb{E}[\exp(\lambda \sum_{t=1}^n m_t)]$$

$$\leq \sqrt{\mathbb{E}\exp(2\lambda \sum_{t=1}^n m_t')}$$

$$\leq \sqrt{\mathbb{E}[\exp(2\lambda^2 \sum_{t=1}^n \|\boldsymbol{\Sigma}_e^{\frac{1}{2}}(\boldsymbol{V} - \boldsymbol{Z})^T \boldsymbol{x}_{t-1}\|_2^2)]}$$

$$= \sqrt{\mathbb{E}\left[\exp\left(2\lambda^2 \text{vec}(\boldsymbol{X}^T)^T (\boldsymbol{I}_n \otimes (\boldsymbol{V} - \boldsymbol{Z})\boldsymbol{\Sigma}_e(\boldsymbol{V} - \boldsymbol{Z})^T)\text{vec}(\boldsymbol{X}^T)\right)\right]}$$

$$= \sqrt{\mathbb{E}\left[\exp\left(2\lambda^2 \boldsymbol{g}^T \boldsymbol{\Upsilon}_x^{\frac{1}{2}} (\boldsymbol{I}_n \otimes (\boldsymbol{V} - \boldsymbol{Z})\boldsymbol{\Sigma}_e(\boldsymbol{V} - \boldsymbol{Z})^T)\boldsymbol{\Upsilon}_x^{\frac{1}{2}} \boldsymbol{g}\right)\right]}, \tag{108}$$

where the first inequality follows Lemma 11 by noting $\{m_t'\}$ is the decoupled sequence tangent to $\{m_t\}$ satisfying $m_t' \sim \mathcal{N}(0, \|\boldsymbol{\Sigma}_e^{\frac{1}{2}}(\boldsymbol{V} - \boldsymbol{Z})^T \boldsymbol{x}_{t-1}\|_2^2)$ conditioned on $\mathcal{F}_{t-1} = \sigma(\{\boldsymbol{e}_0, \cdots, \boldsymbol{e}_{t-1}\})$. The second inequality is from the fact that $\{m_t'\}$ is conditionally independent given $\mathcal{G}$ whose existence is guaranteed by Lemma 10 and the property of sub-Gaussian variables [49, Definition 2.2]. The last equality uses the rotation invariance of Gaussian vectors for $\text{vec}(\boldsymbol{X}^T) \sim \mathcal{N}(\boldsymbol{0}, \boldsymbol{\Upsilon}_x)$ where $\boldsymbol{\Upsilon}_x = \mathbb{E}[\text{vec}(\boldsymbol{X}^T)\text{vec}(\boldsymbol{X}^T)^T]$ and $\boldsymbol{g}$ is a standard Gaussian vector.

We could rearrange the component of (108) further

$$\mathbb{E}\left[\exp\left(2\lambda^2 \boldsymbol{g}^T \boldsymbol{\Upsilon}_x^{\frac{1}{2}} (\boldsymbol{I}_n \otimes (\boldsymbol{V} - \boldsymbol{Z})\boldsymbol{\Sigma}_e(\boldsymbol{V} - \boldsymbol{Z})^T)\boldsymbol{\Upsilon}_x^{\frac{1}{2}} \boldsymbol{g}\right)\right] = \mathbb{E}[\exp(2\lambda^2 \boldsymbol{g}^T \boldsymbol{Q}^T \boldsymbol{Q} \boldsymbol{g})]$$

$$= \mathbb{E}[\exp(2\lambda^2 \boldsymbol{g}^T \boldsymbol{V}_{\boldsymbol{Q}} \boldsymbol{\Sigma}_{\boldsymbol{Q}}^2 \boldsymbol{V}_{\boldsymbol{Q}}^T \boldsymbol{g})]$$

$$= \mathbb{E}[\exp(2\lambda^2 \boldsymbol{z}^T \boldsymbol{\Sigma}_{\boldsymbol{Q}}^2 \boldsymbol{z})]$$

$$= \mathbb{E}[\exp(2\lambda^2 \sum_{i=1}^{nd} s_i^2 z_i^2)], \tag{109}$$

where $U_Q \Sigma_Q V_Q^T$ is the singular value decomposition of $Q = (I_n \otimes \Sigma_e^{\frac{1}{2}}(V - Z)^T)\Upsilon_x^{\frac{1}{2}}$, $\{s_i\}$ are the singular values of $Q$ and $\{z_i\}$ are the entries of $z = V_Q^T g$. Here $z$ is a standard Gaussian vector from the rotation variance of Gaussian vectors and the fact $V_Q$ is an unitary matrix.

From the fact that $\{z_i\}$ are independent standard Gaussian variables, we could bound (109) as

$$\mathbb{E}[\exp(2\lambda^2 \sum_{i=1}^{nd} s_i^2 z_i^2)] \leq \exp(C^2\lambda^2 \sum_{i=1}^{nd} s_i^2) \leq \exp(C^2\lambda^2\|Q\|_{\mathrm{F}}^2), \tag{110}$$

for $\lambda \leq \frac{1}{C\|Q\|}$ where $C$ is a positive constant. Here the first inequality uses the property of sub-Gaussian variables [42, Proposition 2.5.2] and the second inequality is from the definition of $\|Q\|_{\mathrm{F}}$.

Combining (108), (109) and (110), we could bound the MGF of $\langle V - Z, X^T E \rangle$ as

$$\mathbb{E}\Big[ \exp\Big(\lambda \sum_{t=1}^{n} e_t^T (V - Z)^T x_{t-1}\Big)\Big] \leq \sqrt{\mathbb{E}\exp(2\lambda \sum_{t=1}^{n} m_t')}$$

$$\leq \sqrt{\mathbb{E}[\exp(2\lambda^2 \sum_{i=1}^{nd} s_i^2 z_i^2)]}$$

$$\leq \exp(\frac{1}{2}C^2\lambda^2\|Q\|_{\mathrm{F}}^2), \tag{111}$$

for $\lambda \leq \frac{1}{C\|Q\|}$.

After bounding the MGF, we could derive the tail bound

$$P\Big(\langle V - Z, X^T E \rangle > u\Big)$$

$$\leq \frac{\mathbb{E}\Big[ \exp\Big(\lambda\langle V - Z, X^T E \rangle\Big)\Big]}{\exp(\lambda u)}$$

$$\leq \exp\Big( -\min(\frac{u^2}{2C^2\|Q\|_{\mathrm{F}}^2}, \frac{u}{2C\|Q\|})\Big)$$

$$\leq \exp\Big( -\min(\frac{u^2}{2C^2 n\|\Upsilon_x\|\|\Sigma_e\|\|V - Z\|_{\mathrm{F}}^2}, \frac{u}{2C\|\Upsilon_x\|^{\frac{1}{2}}\|\Sigma_e\|^{\frac{1}{2}}\|V - Z\|_{\mathrm{F}}})\Big), \tag{112}$$

where the second inequality is from (111) and the choice of $\lambda$ to minimize the quadratic function. The third inequality holds by noting $\|Q\|_{\mathrm{F}}^2 \leq n\|\Upsilon_x\|\|\Sigma_e\|\|V-Z\|_{\mathrm{F}}^2$ and $\|Q\| \leq \|\Upsilon_x\|^{\frac{1}{2}}\|\Sigma_e\|^{\frac{1}{2}}\|V-Z\|_{\mathrm{F}}$, where we use the facts $\|AB\|_{\mathrm{F}} \leq \|A\|\|B\|_{\mathrm{F}}$, $\|AB\| \leq \|A\|\|B\|$ and $\|A\| \leq \|A\|_{\mathrm{F}}$ for two matrices $A$ and $B$.

By considering the coefficient $1/n$, we could rearrange (112) as

$$P\Big(\frac{1}{n}\langle V - Z, X^T E \rangle > u\Big)$$

$$\leq \exp\Big( -\min(\frac{u^2}{2C^2\frac{1}{n}\|\Upsilon_x\|\|\Sigma_e\|\|V - Z\|_{\mathrm{F}}^2}, \frac{u}{2C\frac{1}{n}\|\Upsilon_x\|^{\frac{1}{2}}\|\Sigma_e\|^{\frac{1}{2}}\|V - Z\|_{\mathrm{F}}})\Big)$$

$$\leq \exp\Big( -\min(\frac{u^2}{2C^2\frac{1}{n}\kappa_{\max}\|\Sigma_e\|\|V - Z\|_{\mathrm{F}}^2}, \frac{u}{2C\frac{1}{n}\kappa_{\max}^{\frac{1}{2}}\|\Sigma_e\|^{\frac{1}{2}}\|V - Z\|_{\mathrm{F}}})\Big), \tag{113}$$

where the second inequality is due to Assumption 2.

The other direction of the tail bound follows the same procedure.

In this way, we illustrate the increment $\frac{1}{n}\langle V - Z, X^T E \rangle$ has a mixed tail with $d_2(\cdot) = \sqrt{2}C\sqrt{\kappa_{\max}}\|\Sigma_e\|^{\frac{1}{2}}\|\cdot\|_{\mathrm{F}}/\sqrt{n}$ and $d_1(\cdot) = 2C\sqrt{\kappa_{\max}}\|\Sigma_e\|^{\frac{1}{2}}\|\cdot\|_{\mathrm{F}}/n$. Following the procedure

in the proof of Lemma 4 to deal with $\gamma_2$-functional and $\gamma_1$-functional, we could rearrange the following formula under the condition of $n > (\omega(\mathcal{C} \cap \mathbb{S}_F) + u)^2$

$$
\begin{aligned}
&\gamma_2(T, d_2) + \gamma_1(T, d_1) + u\Delta_2(T) + u^2\Delta_1(T) \\
&\leq \frac{\sqrt{2}CC'\sqrt{\kappa_{\max}}\|\boldsymbol{\Sigma}_e\|^{\frac{1}{2}}}{\sqrt{n}}\omega(\mathcal{C} \cap \mathbb{S}_F) + \frac{2C(C')^2\sqrt{\kappa_{\max}}\|\boldsymbol{\Sigma}_e\|^{\frac{1}{2}}}{n}\omega(\mathcal{C} \cap \mathbb{S}_F)^2 \\
&\quad + 2\frac{\sqrt{2}C\sqrt{\kappa_{\max}}\|\boldsymbol{\Sigma}_e\|^{\frac{1}{2}}}{\sqrt{n}}u + \frac{4C\sqrt{\kappa_{\max}}\|\boldsymbol{\Sigma}_e\|^{\frac{1}{2}}}{n}u^2 \\
&\leq C''\sqrt{\kappa_{\max}}\|\boldsymbol{\Sigma}_e\|^{\frac{1}{2}}\Big(\frac{\omega(\mathcal{C} \cap \mathbb{S}_F) + u}{\sqrt{n}} + \frac{\omega(\mathcal{C} \cap \mathbb{S}_F)^2 + u^2}{n}\Big) \\
&\leq C''\sqrt{\kappa_{\max}}\|\boldsymbol{\Sigma}_e\|^{\frac{1}{2}}\Big(\frac{\omega(\mathcal{C} \cap \mathbb{S}_F) + u}{\sqrt{n}} + \frac{(\omega(\mathcal{C} \cap \mathbb{S}_F) + u)^2}{n}\Big) \\
&\leq 2C''\sqrt{\kappa_{\max}}\|\boldsymbol{\Sigma}_e\|^{\frac{1}{2}}\frac{\omega(\mathcal{C} \cap \mathbb{S}_F) + u}{\sqrt{n}},
\end{aligned}
\tag{114}
$$

where the last inequality is from the condition of $n > (\omega(\mathcal{C} \cap \mathbb{S}_F) + u)^2$.

Combining with Lemma 6, we could derive

$$
P\Big(\sup_{\boldsymbol{V} \in \mathcal{C} \cap \mathbb{S}_F} |\frac{1}{n}\langle \boldsymbol{V}, \boldsymbol{X}^T\boldsymbol{E}\rangle| > C'''\sqrt{\kappa_{\max}}\|\boldsymbol{\Sigma}_e\|^{\frac{1}{2}}\frac{\omega(\mathcal{C} \cap \mathbb{S}_F) + u}{\sqrt{n}}\Big) \leq 2\exp(-u^2),
\tag{115}
$$

under the condition of $n > (\omega(\mathcal{C} \cap \mathbb{S}_F) + u)^2$.

## D Additional numerical results

### D.1 Background modeling

We evaluate the result in Corollary 2 through the background modeling problem in [50]. The background modeling problem aims to reconstruct the static background through a sequence of video frames with moving objects in the foreground. By vectoring and stacking the frames as columns of the observed matrix $\boldsymbol{Y}$, the static background is modeled as a low-rank component $\boldsymbol{L}_\star$ and the foreground is viewed as a dynamic and sparse perturbation $\boldsymbol{S}_\star$. In this way, the background modeling problem could be viewed as the robust PCA problem considered in Corollary 2 without noise. We use the first 1600 frames of the *Highway* dataset in [50] at a resolution of $320 \times 240$. By dividing the video into parts with 200 frames, we select the constraint parameters in (29) through cross validation and then solve the problem by AltPGD (Algorithm 2). Figure 5(a) and 5(b) show one original frame and its extracted background from the video frames.

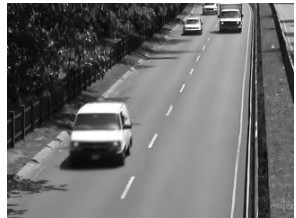 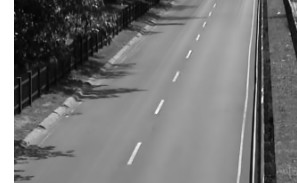

    (a) Original input frame               (b) Low-rank frame

Figure 5: Background modeling in the *Highway* video.