# OpenReview forum: "Linear Convergence of Gradient Methods for Estimating Structured Transition Matrices in High-dimensional Vector Autoregressive Models"
_NeurIPS.cc/2021/Conference — NeurIPS 2021 Poster_

### Official Review · Reviewer_cNh8 · 2021-07-13

**Rating:** 7
**Confidence:** 4

**Summary:**

This manuscript proposes two algorithms for estimation of transition matrices in single-structured as well as superposition-structured VAR models. Linear convergence to true parameters are verified under certain stability conditions, and additional assumptions on the data distribution, step size and minimum sample size. Some empirical results are reported to illustrate good performance of proposed algorithms in finite sample cases.

**Main Review:**

- On results of Theorem 1: The consistency rate appeared in equation 16 and then later on equation 18, seems to depend on the some norm of the covariance of error terms (It is not clear which norm is used since authors have not defined the notation in the manuscript). This norm may be related to the dimensions of the covariance matrix of error terms and might be at least of order d. This will impose an upper bound on d, at least of the form $d \leq n$ which makes it impossible for the results to hold in high-dimensional cases in which $d > n$. Further, if one includes the bound in equation 15, the consistency rate in equation 16 might include $\sqrt{d}$ which does not even vanish with the sample size. The authors are encouraged to elaborate further on the consistency rate in Theorem 1 (similarly in Theorem 2), specifically discuss under which combinations of n and d, the consistency rates would vanish in their theoretical results.


- Assumption 3 is a rather technical condition and hard to check given a specific superposition-structured VAR model. It would be appropriate for the authors to simplify this assumption at least for some specific superposition-structured cases such as the low rank plus sparse one. Moreover, it is not clear whether this assumption ensures the identifiability of transition matrices L and S. The identifiability issue is only mentioned in Remark 6, but not discussed further. This is a serious issue which should be addressed thoroughly, otherwise the interpretation of estimates of transition matrices would be impossible. There are results in the literature which could help verifying the identifiability in some specific cases, such as Theorem 1 in Hsu et. al. (2011) for the case of low rank plus sparse.



- The manuscript seems to be written hastily since there are some incomplete sentences and several mathematical notations which are either not defined at all or defined after the first time being used in the manuscript which might be confusing for readers. Some examples:

- Sentence seems to be incomplete: page 2, line 72, ``when the transition ..."

- Incomplete sentence on page 3, line 86: ``we require ..."

- cone(.) is not defined but used on page 4, line 108

- The set $S_F$ used in equation 15, page 4, but not defined before (Same for $S_2$ in equation 35 in the supplement).

- Which norm on the covariance of error terms does appear in equation 18? Is it the operator norm?

- The notation script P, i.e. $\mathcal{P}$ (projection) is used in equation 5 on page 3, but not defined before.






- Gaussianity seems to be a major assumption throughout the manuscript. This assumption is restrictive in applications. The authors need to discuss whether the results hold for some heavier tail distributions, and if not, describe the main theoretical bottlenecks.



- On the algorithms: (1) Is it possible/needed to select different step sizes for the superposition-structured case in equation 21? (2) How are the projections performed in both algorithms 1 and 2 for the low rank and sparse components? This needs to be added for completeness; (3) There is no discussion on how step sizes are selected/computed in numerical studies. Convergence in both algorithms might be sensitive to this hyperparameter as hinted by the theoretical results (there is a lower bound requirement for step size in all theoretical results in the manuscript). This issue should be addressed comprehensively.



- Real data: There is no comparison with competing methods such as the FNSL in the real data section, i.e. section 5.2 (similarly in the additional numerical results in section D in the supplement).

- It is not clear why the inequality in equation 18 holds, please elaborate.

Reference:

Hsu, D., Kakade, S. M., \& Zhang, T. (2011). Robust matrix decomposition with sparse corruptions. IEEE Transactions on Information Theory, 57(11), 7221-7234.

**Time Spent Reviewing:**

around 8

---

> ### Author Response · Authors · 2021-08-10
> **Response to Reviewer cNh8**
>
> We thank the reviewer for the detailed feedback and insights.
>
> **Q1: Clarify the norm for the covariance of error terms and the consistency rate.**
> **A1:** Thanks for pointing out this problem. The norm for the covariance of error terms is the spectral norm. The spectral norm is not related to the dimension of the covariance matrix of error terms for common applications. For example, the covariance matrix of noise in [1] is the identity matrix multiplied by a scalar. So the spectral norm of the covariance of error terms does not affect the consistency rate in Equation 16 and 18 for common applications.
>
> **Q2: Interpretation of the identifiability.**
> **A2:** In Section 3 of [2], the authors give an interpretation for the low rank plus sparse case when the covariance matrix is an identify matrix. In this case, the maximum singular value of the product of projection matrices measures the “angle” between the subspaces. The structural incoherence condition imposes that the subspaces where the sparse matrix and the low-rank matrix live would be sufficiently “orthogonal” to each other. The identifiability is ensured in the sense that a portion of the sparse matrix (low-rank matrix) could not be well approximated by adding a small number of additional terms to the low-rank matrix (sparse matrix). Thus the two matrices could not be sparse and low-rank simultaneously.
> On the other hand, when we consider the model with independent samples and the noiseless case, the sum of two Frobenius norms in Equation 25 would converge to zero, under Assumption 3. This means we could estimate the two matrices exactly and identify the sum of two matrices.
>
> **Q3: The main theoretical bottlenecks to extend the Gaussian distribution assumption.**
> **A3:** When we prove Lemma 4 and Lemma 5, we refer to the Hanson-Wright inequality [3] to derive the concentration inequality for the chaos in Equation 90. The Hanson-Wright inequality requires the random vector has independent sub-Gaussian entries and this requirement could be satisfied by the Gaussian distribution assumption. If we consider some heavier tail distributions, new techniques are required to deal with the chaos. It would be an interesting direction for future works.
>
> **Q4: Implementation of the algorithm.**
> **A4:** First, the step size for the superposition-structured case is also selected as the inverse of the maximum eigenvalue of the covariance matrix of the samples in Theorem 2. In fact, smaller step size could also lead to the linear convergence rate. The choice in the manuscript is for the brevity of the theorems. Second, we promote the sparsity by the l1-norm and promote the low-rank structure by the nuclear norm as Equation 28. We follow the procedure in [4] to perform the projection for the sparse component. And the projection for the low-rank component is a union of the singular value decomposition and the projection for the sparse component, which is applied to the vector composed of the singular values. We would add this description in the final version. Thirdly, we choose the step size as the inverse of the maximum eigenvalue of the estimated covariance matrix of the samples in simulations. The estimation of the covariance is derived from $X^T X / n$, where $X$ is the data matrix.
>
> **Q5: Other algorithms for the simulation with real data.**
> **A5:** In fact, we also try to use FNSL in the simulation of Section 5.2. However, with the parameter selected through 5-fold cross validation, there would be few entries in the estimated transition matrix with huge values and the other non-zero entries only have tiny values. This result is not as interpretable as the one of PGD, so we do not include it in the manuscript. For the simulation of robust PCA in Section D, there are several efficient algorithms special for this application. Our simulation result is only to verify that our analysis framework also adapts to this model, so we do not compare with other algorithms.
>
> **Q6: Derivation of Equation 18.**
> **A6:** The obscurity of Equation 18 might be from the definition of $\xi$, which is placed in Equation 43 (in the supplementary material). The first equality in Equation 18 is based on this definition. If we apply the inequality about $\rho$ in Equation 17 to the left hand of Equation 18, we could derive the final inequality. We would move the definition of $\xi$ to Theorem 1 to make the theorem more interpretable in the final version.
>
> **Q7: Incomplete sentences and undefined notations.**
> **A7:** Thank you for pointing out these. We would correct these sentences and add the relative definitions of notations in the final version. Here “cone(A)” represents the cone generated by the set A. $S_{F}$ and $S_{2}$ represent the sphere with unit Frobenius norm and the unit Euclidean sphere separately.
>
> [1] M. Simchowitz, H. Mania, S. Tu, M. I. Jordan, and B. Recht. Learning without mixing: Towards a sharp analysis of linear system identification. Proceedings of the 31st Conference On Learning Theory, vol. 75 of Proceedings of Machine Learning Research, pp. 439–473, 2018.
> [2] K. Greenewald and A. O. Hero. Robust Kronecker product PCA for spatio-temporal covariance estimation. IEEE Transactions on Signal Processing, vol. 63, no. 23, pp. 6368–6378, 2015.
> [3] M. Rudelson and R. Vershynin. Hanson-Wright inequality and sub-gaussian concentration. Electronic Communications in Probability, vol. 18, 2013.
> [4] J. Duchi, S. Shalev-Shwartz, Y. Singer, and T. Chandra. Efficient projections onto the l1-ball for learning in high dimensions. Proceedings of the 25th International Conference on Machine Learning, 2008.

---

### Official Review · Reviewer_Ptje · 2021-07-16

**Rating:** 6
**Confidence:** 3

**Summary:**

The paper proposes projected gradient descent algorithms for estimating high-dimensional vector auto-regressive models, and conducts convergence analysis following the restricted strong convexity lens.  Specifically, sparse, low-rank and sparse plus low-rank models are of interest.  While many of these topics have been studied "to death", the novelty here seems to be (a) there's limited work in applying sparse plus low-rank approaches to time-series settings, where there is an additional constraint that the transition matrix has to lead to a stable dynamical system (b) Typical convergence analysis is done under iid setting, but for VAR models, samples are naturally dependent with strong auto-correlation.

**Main Review:**

Overall it's an interesting paper, but I'm having a difficult time judging its originality,  as each individual piece has been proposed and explored in numerous papers,  but the particular combination of ideas considered in this paper (using RSC arguments to analyze PGD methods for sparse+low-rank problem for the VAR setting and establishing linear convergence) appears to be new.   Many of the new tools + challenges of applying sparse + low-rank methods to VAR setting (e.g. use of spectral densities, e.t.c) have been addressed in earlier publications (e.g. [22]), analysis of non-iid data, and applications of RSC arguments, are also fairly well known, the authors provide a good overview in the intro.   So I'm not sure if the authors are filling-in a natural gap in a tensor of cross-products of various ideas,  or if there's indeed some key novelty in this paper.  Perhaps the authors can comment / highlight.

The main limitations of the paper:
1) There is a wealth of algorithms for solving high-dimensional sparse problems, and to my knowledge PGD methods are not  among the most competitive.  I would like the authors to comment on e.g. randomized block-coordinate descent methods, accelerated proximal methods, and all the other popular solvers (where most of these have been applied to sparse/group-sparse/sparse+low-rank, e.t.c problem settings).   Adding a projection (rescaling) to reduce spectral radius for VAR setting should be applicable to many of these.
Baseline comparison is done with only one method, "FNSL",  which seems to be of accelerated proximal gradient type,  and where parameters are not chosen properly (so there's a typical oscillation pattern).   It'd be more convincing if there were other baselines.

2) Evaluation on real data is very limited, and the analysis of 'financial data' is naive and misleading / wrong.  VAR model is build on log-prices,  which are non-stationary,  to make sense the paper should apply the analysis to log-returns.    Finding correlations isn't very interesting -- one can use a sample covariance matrix and clustering for that -- if you can comment on interesting lead-lag relationships that would be more interesting.

Minor comments:
1) It's unclear what is "connectivity measuring among financial firms",  and also it's unclear why radar/SAR signal processing would have a network structure -- is there perhaps a low-rank spatial structure? Please explain clearly.
2) 'studied by numerous literatures' - awkward.
3) Please take a look at references for some clean-up e.g. pca->PCA .. S&p -> S&P, e.t.c. , ref[40], ...



**Time Spent Reviewing:**

3

---

> ### Author Response · Authors · 2021-08-10
> **Response to Reviewer Ptje**
>
> We thank the reviewer for providing detailed suggestions.
>
> **Q1: Comparison with the existing works and novelty of our work.**
> **A1:** We first clarify our contribution in this article. The existing works on structured VAR models (e.g., [1], [2], and [3]) are mainly concerned with establishing the statistical error bounds for different recovery procedures. These results are known as structure-data tradeoffs, which identify how many samples (or data) can successfully recover a structured signal for a given recovery procedure. The current work focuses on time-data tradeoffs, which reveal how many samples (or data) can ensure that an iterative algorithm (used to solve a given recovery procedure) achieves a fast convergence rate (or saving time). On the other hand, our work could be regarded as a generalization of the result in [4] from independent samples to time series. This generalization, however, is highly nontrivial, because we need to develop new mathematical tools to address time-series settings (e.g., two deviation inequalities: Lemmas 4 and 5). To the best our knowledge, Lemma 5 seems to be the first non-asymptotic result for the VAR models which yields the unified estimation error bounds for both independent and correlated samples. So our results can be regarded as studying data-time tradeoffs in structured VAR problems.
>
> **Q2: Discussion about the baseline comparison.**
> **A2:** Thanks for this helpful suggestion. When the loss function is not strongly convex, traditional convex optimization literature predicts that the gradient-based algorithms (e.g., PGD, randomized block-coordinate descent methods, and accelerated proximal methods) cannot achieve a linear convergence rate. However, when the structural information of the desired signal is incorporated into the iterations, the situation will change. For example, in PGD (or randomized block-coordinate descent methods), the projection makes each iteration well structured, which enables us to consider the constrained singular values of certain matrices and to achieve the linear convergence rate. For accelerated proximal methods, it seems unclear to integrate the structural information into the iterations. Actually, FNSL can be regarded as an accelerated proximal method, which achieves the sublinear convergence rate.
> To the best of our knowledge, [1] is the only literature which explores the efficient algorithm and its convergence rate to solve structured VAR models. So we choose FNSL proposed in [1] as the baseline. FNSL is based on the Nesterov’s acceleration of the proximal gradient descent and the oscillation is a basic phenomenon of the Nesterov’s acceleration. The oscillation phenomenon is presented in Figure 10.1 in [5] and is illustrated theoretically through the ODE in [6]. For FNSL, we refer to the grid search around $\sqrt{n \log d}$ to choose the parameter $\lambda$ in the penalized model and this choice is guaranteed theoretically by Proposition 1 in [1].
>
> **Q3: Discussion about the simulation with real data.**
> **A3:** First, both prices and returns of stocks are widely used in the analysis of VAR models. For example, [7], [8], [9] apply log-prices or prices in the simulations and [1], [10] apply log-returns. The main goal of our simulation is to predict the stock prices, so we choose the log-prices. The covariance estimation and the clustering cannot complete the prediction task. When estimating the VAR model, we find the interesting phenomenon that the future prices of stocks are affected in opposite manners by the current prices of stocks in the same factor and the ones in different factors. This phenomenon coincides with the reality. The lead-lag effect could be used to illustrate the relationship between the price movements of two markets. And the study of the lead-lag effect is often based on the VAR models, see e.g., [10]. So it would be an interesting direction to explore this in future works.
>
> **Q4: Typos.**
> **A4:** Thanks for pointing out these typos. We would correct them in the final version.
>
> [1] S. Basu, X. Li, and G. Michailidis. Low rank and structured modeling of high-dimensional vector autoregressions. IEEE Transactions on Signal Processing, vol. 67, pp. 1207–1222, 2019.
> [2] S. Basu and G. Michailidis. Regularized estimation in sparse high-dimensional time series models. The Annals of Statistics, vol. 43, pp. 1535–1567, 2015.
> [3] I. Melnyk and A. Banerjee. Estimating structured vector autoregressive models. International Conference on Machine Learning, pp. 830–839, 2016.
> [4] S. Oymak, B. Recht, and M. Soltanolkotabi. Sharp time–data tradeoffs for linear inverse problems. IEEE Transactions on Information Theory, vol. 64, no. 6, pp. 4129–4158, 2018.
> [5] A. Beck, First-order methods in optimization. Society for Industrial and Applied Mathematics, 2017.
> [6] W. Su, S. Boyd, and E. Candès. A differential equation for modeling Nesterov’s accelerated gradient method: theory and insights. Journal of Machine Learning Research, 2016.
> [7] F. Huang and S. Chen. Learning dynamic conditional gaussian graphical models. IEEE Transactions on Knowledge and Data Engineering, vol. 30, pp. 703–716, 2018.
> [8] F. Han and H. Liu. Transition matrix estimation in high dimensional time series. Proceedings of the 30th International Conference on Machine Learning, pp. 172–180, 2013.
> [9] F. Han, H. Lu, and H. Liu. A direct estimation of high dimensional stationary vector autoregressions. Journal of Machine Learning Research, vol. 16, no. 97, pp. 3115–3150, 2015.
> [10] J. Lin and G. Michailidis. Regularized estimation of high-dimensional factor-augmented vector autoregressive (FAVAR) models. Journal of Machine Learning Research, vol. 21, no. 117, pp. 1–51, 2020.

---

> > ### Comment · Reviewer_Ptje · 2021-08-13
> > **reply to authors comments**
> >
> > Thank you for the response! I appreciate pointing out the novelty in lemmas 4 and 5, it's worth making the contribution clearer in the paper.
> >
> > "The oscillation phenomenon is presented in Figure 10.1 in [5] and is illustrated theoretically through the ODE in [6]."
> > There are effective practical approaches to reduce oscillations in Nesterov-style accelerated schemes by using restarts, although I'm less confident about their theoretical guarantees, e.g. see:
> > https://statweb.stanford.edu/~candes/teaching/math301/Lectures/adap_restart_nesterov.pdf
> >
> > Regarding using {log-}prices vs. their differences -- if there's no mean-reversion (unusual for individual stocks), under the martingale assumption y(t+1) = y(t) + stationary bit... , so the behavior is inherently non-stationary ({geometric} brownian motion) with variance growing linearly,  contradicting the assumptions of the paper (on the spectrum of the transition matrix).  I'm sure there are papers in the literature that may be cavalier about real data experiments, but I just wanted to clarify what you're doing, and if you're not using stationary time-series, how do you reconcile it with the assumptions of your paper.

---

> > > ### Author Response · Authors · 2021-08-16
> > > **Response to Reviewer Ptje**
> > >
> > > **Clarify our contribution.**
> > > **A1:** Thanks for this helpful suggestion. We will make our contribution clearer in the revised version.
> > >
> > > **Nesterov-style accelerated schemes with restarts.**
> > > **A2:** Thank you for this further comment. In the first response, we only consider the original Nesterov-style accelerated schemes (without restarts), which have shown an oscillation pattern in practices. To avoid this oscillation, restarting the algorithm at non-monotonicity is suggested in [1] and [2]. For strongly convex objective functions, the linear convergence rate of these accelerated schemes with restarts is first established in [1]. For general convex objective functions, only the sub-linear rate is developed in [2]. In the current paper, the objective function is not strongly convex, so we have integrated the structural information of the desired signal into the iterations of PGD to derive a linear rate. It would be an exciting direction to explore the linear convergence rate of accelerated schemes with restarts for structured VAR problems.
> > >
> > > **The non-stationary property of log-prices.**
> > > **A3:** Thanks for this critical comment which has provided us a more in-depth understanding of the stationary time series in practical applications. The non-stationary time series might cause the spurious regression, which would indicate spurious relationships among variables. So we follow this suggestion and have performed the similar simulations with the log-returns of stocks. We will present the new numerical results (with sparse pattern of the estimated transition matrix) in the revised version.
> > >
> > > [1] B. O’Donoghue and E. Candès. Adaptive restart for accelerated gradient schemes. Foundations of Computational Mathematics, vol. 15, pp. 715–732, 2013.
> > > [2] P. Giselsson and S. Boyd. Monotonicity and restart in fast gradient methods. 53rd IEEE Conference on Decision and Control, pp. 5058–5063, 2014.

---

> > > > ### Comment · Reviewer_Ptje · 2021-08-22
> > > > **reply to authors**
> > > >
> > > > Thank you, that sounds good, I increased the rating.

---

### Official Review · Reviewer_ZMT4 · 2021-07-16

**Rating:** 6
**Confidence:** 2

**Summary:**

This paper considers the problem of estimating the transition matrix of a lag-1 vector autoregressive model. The authors propose projected gradient descent as well as alternating projected gradient descent algorithms to learn both single- and supervision-structured transition matrices, respectively. Non-asymptotic analysis is also provided which shows the algorithms convergence linearly tp the statistical error in the high-dimensional settings.

**Limitations And Societal Impact:**

- The paper focuses on the special case of VAR models of lag 1 with serially uncorrelated Gaussian errors.
- The propose methods assume that the transition matrix is single-structured or a sum of two single-structured components. This assumption could be restrictive for real data.


**Main Review:**

The paper is well-written and the theoretical results look solid. The authors propose algorithms for learning a single-structured transition matrix as well as a transition matrix as the sum of two single-structured components. The theoretical analysis addresses the key challenge that data matrix has dependent rows, using the extreme eigenvalue bounds for the covariance matrix of Gaussian processes presented in S. Basu  et al. 2015. The authors prove sharp convergence guarantees for the proposed algorithms. Numerical results on S&P500 stock prices are provided to support the theoretical analysis.

**Time Spent Reviewing:**

2

---

> ### Author Response · Authors · 2021-08-10
> **Response to Reviewer ZMT4**
>
> Thank you very much for your valuable feedback.
>
> **Q1: Discussion on restricted assumptions about the models.**
> **A1:** In this paper, our discussion focuses on the VAR(1) model for simplification. Indeed, as discussed in Remark 4, the VAR(d) model with arbitrary d could be reformulated as a VAR(1) model as in [1] and [2]. Thus our analysis for the VAR(1) model can naturally adapt to the VAR(d) model.
> The assumption of serially uncorrelated Gaussian errors is required in the proofs of Lemmas 4 and 5. However, this assumption is common in literature (see, e.g., [1] and [3]). In the proof, we have utilized the Hanson-Wright inequality [4] to derive the concentration inequality for the chaos in Equation 90. The Hanson-Wright inequality requires the random vector has independent sub-Gaussian entries, which is satisfied by the Gaussian distribution assumption. It would be an interesting future direction to extend the analysis to general distributions.
> In this paper, we only discuss the single structure or the sum of two structures for simplification. As discussed in [5], the structural incoherence can be defined for the models with any number of structures. If we adjust the description of Assumption 3 for any number of structures as [5], our analysis would also adapt to the general condition.
>
> [1] S. Basu and G. Michailidis. Regularized estimation in sparse high-dimensional time series models. The Annals of Statistics, vol. 43, pp. 1535–1567, 2015.
> [2] I. Melnyk and A. Banerjee. Estimating structured vector autoregressive models. International Conference on Machine Learning, pp. 830–839, 2016.
> [3] S. Basu, X. Li, and G. Michailidis. Low rank and structured modeling of high-dimensional vector autoregressions. IEEE Transactions on Signal Processing, vol. 67, pp. 1207–1222, 2019.
> [4] M. Rudelson and R. Vershynin. Hanson-wright inequality and sub-gaussian concentration. Electronic Communications in Probability, vol. 18, 2013.
> [5] E. Yang and P. K. Ravikumar. Dirty statistical models. Advances in Neural Information Processing, 2013.

---

> > ### Comment · Reviewer_ZMT4 · 2021-08-29
> > **After rebuttal**
> >
> > Dear authors, thank you for your response. I've read the response and I am keeping my score as it is.

---

### Official Review · Reviewer_CH7C · 2021-07-17

**Rating:** 6
**Confidence:** 4

**Summary:**

The paper analyzes convergence of gradient descent algorithm for estimating the transition matrix under vector-autoregressive models under (i) high-dimensional regularization assumptions on the matrix and (ii) a sparse+low-rank kind of complementary regularization structure on the matrix. The results are more or less along expected lines with one technical difficulty to overcome in the proof (which is done in Lemma 5).

**Ethical Concerns:**

none (to this reviewers knowledge)

**Limitations And Societal Impact:**

none (to this reviewers knowledge)

**Main Review:**

1) The model and the setup considered are general high-dimensional setup and superposition setup. However, I think it would also be good to also consider and discuss the case when there is no regularization (I.e., the low-dimensional setup). I suspect the Projection makes the iterates bounded which enables the analysis at several places. However, in the low-dimensional setup without any projection, the unboundedness of the iterates might cause issues for the analysis ? A discussion on this would be welcome.

2) If we remove all the standard (i.e., by now well-known steps), the main contribution is in Lemma 5. It might be worth discussing this method in the main paper.

3) Theorem 2 and Corollary 3 from [18] also provides optimization error. This should be discussed.

4) Minor: the term 'sharp' is typically referred to the case when the constants also match some form of lower bounds. If upper and lower bound match up to constants, it is typically referred to as 'rate-optimal' 1

==========
Thanks for the response. I maintain my score.

To me the more interesting problem to consider is to analyze the dynamics of the algorithm in the 'real' high-dimensional setup (where the dimension and sample size go to infinity with their ratio being bounded and no structural assumptions), which might be more challenging.


**Time Spent Reviewing:**

10 hours

---

> ### Author Response · Authors · 2021-08-10
> **Response to Reviewer CH7C**
>
> We appreciate the constructive feedback and valuable suggestions.
>
> **Q1: Discussion in the low-dimensional setup.**
> **A1:** Thanks for this insightful comment. In the high-dimensional setting, the projection plays a key role in our analysis. Because it makes each iteration well structured (or bounded), which enables us to consider the constrained singular values of certain matrices and to achieve the linear convergence rate. In the low-dimensional setting (or there is no regularization), each iteration might not be well structured, and our preliminary simulations show that gradient-based algorithms may not achieve a linear convergence rate.
>
> **Q2: Discussion of Lemma 5.**
> **A2:** The contribution of Lemma 5 is twofold. First, to overcome the dependency among samples, we introduce some new tools (e.g., the decoupling theory [1]) in the proof of Lemma 5. Second, Lemma 5 yields the unified estimation error bounds for both independent and correlated samples. The unified form about the non-asymptotic analysis of the VAR models is absent in the literatures. We would add this discussion in the final version.
>
> **Q3: Related literature about the optimization error.**
> **A3:** Thank you for pointing out the related work [2]. In [2], the authors illustrate that PGD enjoys a linear convergence rate to the statistical error for the VAR (1) model with a sparse transition matrix. Compared with the result in [2], our analysis is different in three aspects. First, the spectral norm of the transition matrix in [2] needs to be strictly less than one. This requirement is much more restrictive than Assumption 1 in our analysis (also illustrated in [3]). Second, the analysis in [2] is based on the sparse parameter, while our results adapt to general structured signals. Third, our analysis relies on different techniques, such as the spectral density and the decoupling theory, to allow for milder conditions. We would add this discussion in the final version.
>
> [1] V. H. de la Peña. A general class of exponential inequalities for martingales and ratios. The Annals of Probability, vol. 27, pp. 537–564, 1999.
> [2] P.-L. Loh and M. J. Wainwright. High-dimensional regression with noisy and missing data: Provable guarantees with nonconvexity. The Annals of Statistics, vol. 40, pp. 1637–1664, 2012.
> [3] S. Basu and G. Michailidis. Regularized estimation in sparse high-dimensional time series models. The Annals of Statistics, vol. 43, pp. 1535–1567, 2015.

---

> ### Author Response · Authors · 2021-08-29
> **Response to Reviewer CH7C**
>
> Thanks for this constructive suggestion which provides a valuable research direction for future work. In the previous response, we have mentioned that in the high-dimensional underdetermined cases (the number of measurements is less than the ambient dimension), gradient-based algorithms might not achieve a linear convergence rate if we do not utilize any structural priori. However, it is worth noting that our theoretical result (Theorem 1) also sheds some light on the unstructured case (i.e., without structural assumptions). Indeed, when there is no structural assumption, the descent cone in Theorem 1 becomes the whole space, and hence the required samples to guarantee the linear convergence would be of the order of $\mathcal{d^2}$ (which requires the problem to be overdetermined). This result is also consistent with the traditional optimization theory since the objective function is strongly convex when the samples are enough.

---

### Official Review · Reviewer_SbHr · 2021-07-17

**Rating:** 7
**Confidence:** 4

**Summary:**

This paper presents a recovery guarantee for gradient-based optimization method applied to high-dimensional vector autoregressive (VAR) models. It considers both single-structured and superposition-structured transition matrix, and shows that the distance between the true parameter and the iterates generated by projected gradient descent (PGD) algorithm will decrease linearly to certain statistical error controlled by sample size and Gaussian width of the assumed structure. In the experiments, PGD is shown to outperform the baseline algorithm FNSL empirically.

**Limitations And Societal Impact:**

Yes.

**Main Review:**

Originality:

The result presented in this paper seems new in the literature, but there are several existing works that share something in common with this paper. For VAR models with general structures, [1] has analyzed the statistical error for regularized estimators using Gaussian width. For superposition structures, Gaussian-width-based analysis is available for linear models in [2]. The linear convergence of PGD has been established for general-structured linear models in [3]. Given these works, I feel that to some extent this paper is like a hybridization of them, and the linear convergence result here is not too surprising. Therefore I would not say that the paper has a lot of originality.

Quality:

The technical quality of this paper is reasonably high, but not outstanding. To obtain the error bounds for PGD and AltPGD in Section 3 and 4, some effort seems to be required for dealing with the mathematical details, given that VAR models and superposition structures are relatively complicated. The paper also provides additional results on multitask learning and robust PCA. On the other hand, like the comments above for originality, the theoretical results proved in this paper seem to have similar flavor as what existing works have shown, so that the proof ideas can be borrowed in a relatively straightforward manner. By reading the proofs in the supplementary material, it looks to me that the line of attack is more or less standard.

Clarity:

The presentation of this paper is generally clear, and the technical results are well-organized. For the experiments, one thing that is unclear from the description is how the constraint parameter is set for PGD. In both Eq. (3) and (4), the formulations use the information of the ground truth, which is unknown in practical applications. If the constraint parameter is set as suggested in (3) and (4) but the baseline method FNSL has no access to the ground truth, the comparison might be unfair. Could authors clarify on this?

Significance:

Using PGD for estimating VAR models is straightforward from the algorithmic perspective, but the linear-convergence result established in the paper fills in the missing piece of statistical guarantees.

[1] I. Melnyk and A. Banerjee. Estimating Structured Vector Autoregressive Models. In International Conference on Machine Learning, pages 830–839, June 2016.

[2] Q. Gu and A. Banerjee. High dimensional structured superposition models. In Advances in Neural Information Processing Systems, pages 3684-3692, 2016.

[3] S. Oymak, B. Recht, and M. Soltanolkotabi. Sharp time–data tradeoffs for linear inverse problems. IEEE Transactions on Information Theory, vol. 64, no. 6, pp. 4129–4158, 2018.

********************* Post-rebuttal comments *********************
I tend to agree with the author response on Lemma 4 and 5, and after reading other reviews and comments, I would like to increase my score to 7.

**Time Spent Reviewing:**

4

---

> ### Author Response · Authors · 2021-08-10
> **Response to Reviewer SbHr**
>
> Thank you very much for the helpful comments.
>
> **Q1: Comparison with the existing works and novelty of our work.**
> **A1:** The works [1] and [2] have established the statistical error bounds for single-structured and superposition-structured VAR models respectively. These results are known as structure-data tradeoffs, which identify how many samples (or data) can recover a structured signal for a given recovery procedure. The current work focuses on time-data tradeoffs, which reveal how many samples (or data) can ensure that an iterative algorithm (used to solve a given recovery procedure) achieves a fast convergence rate (or saving time). On the other hand, our work could be regarded as a generalization of the result in [3] from independent samples to time series. This generalization, however, is highly nontrivial, because we need to develop new mathematical tools to address time-series settings (e.g., two deviation inequalities: Lemmas 4 and 5). To the best our knowledge, Lemma 5 seems to be the first non-asymptotic result for the VAR models which yields the unified estimation error bounds for both independent and correlated samples. This result theoretically reveals the influence of time series on the convergence rate of PGD in solving VAR problems.
>
> **Q2: The setting of the constraint parameter for PGD.**
> **A2:** In the simulation with real data (Section 5.2), we select the constraint parameter for PGD through 5-fold cross validation. In the simulation with synthetic data (Section 5.1), the constraint parameter for PGD is set by the information of the ground truth. However, it is worth noting that the stability of the convergence results is insensitive to the mismatch of the constrained parameter (see, e.g., Theorem 9 in [3]). For FNSL, we cannot get the optimal parameter $\lambda$ in the penalized model. We only refer to the grid search around $\sqrt{n \log d}$ to choose the parameter (as suggested by Proposition 1 in [4]). Indeed, the fact that PGD achieves a faster convergence rate than FNSL is because the former has utilized more structure information than the latter.
>
> [1] I. Melnyk and A. Banerjee. Estimating structured vector autoregressive models. International Conference on Machine Learning, pp. 830–839, 2016.
> [2] Q. Gu and A. Banerjee. High dimensional structured superposition models. Advances in Neural Information Processing Systems, pp. 3684-3692, 2016.
> [3] S. Oymak, B. Recht, and M. Soltanolkotabi. Sharp time–data tradeoffs for linear inverse problems. IEEE Transactions on Information Theory, vol. 64, no. 6, pp. 4129–4158, 2018.
> [4] S. Basu, X. Li, and G. Michailidis. Low rank and structured modeling of high-dimensional vector autoregressions. IEEE Transactions on Signal Processing, vol. 67, pp. 1207–1222, 2019.

---

### Decision · Program_Chairs · 2021-09-27

**Decision:**

Accept (Poster)

**Comment:**

Given that the authors have satisfactorily addressed the concerns of the reviewers, the consensus of the review committee is that the paper should be accepted for presentation at Neurips. I would like to ask the authors to read the paper carefully one more time and revise all the typos/missed notation definitions including the ones that are pointed out by the reviewers.